# Melt sensitivity of irreversible retreat of Pine Island Glacier

Brad Reed[1,2], J. A. Mattias Green[1], Adrian Jenkins[2], and G. Hilmar Gudmundsson[2]

[1]School of Ocean Sciences, Bangor University, Menai Bridge, UK
[2]Department of Geography and Environmental Sciences, Northumbria University, Newcastle, UK

**Correspondence:** Brad Reed (brad.reed@northumbria.ac.uk)

**Abstract.** In recent decades, glaciers in the Amundsen Sea Embayment in West Antarctica have made the largest contribution to mass loss from the entire Antarctic Ice Sheet. Glacier retreat and acceleration have led to concerns about the stability of the region and the effects of future climate change. Coastal thinning and near-synchronous increases in ice flux across neighbouring glaciers suggest that ocean-driven melting is one of the main drivers of mass imbalance. However, the response of individual
glaciers to changes in ocean conditions varies according to their local geometry. One of the largest and fastest flowing of these glaciers, Pine Island Glacier (PIG), underwent a retreat from a subglacial ridge in the 1940s following a period of unusually warm conditions. Despite subsequent cooler periods, the glacier failed to recover back to the ridge and continued retreating to its present-day position. Here, we use the ice-flow model Úa to investigate the sensitivity of this retreat to changes in basal melting. We show that a short period of increased basal melt was sufficient to force the glacier from its stable position on
the ridge and undergo an irreversible retreat to the next topographic high. Once high melting begins upstream of the ridge, only near-zero melt rates can stop the retreat, indicating a possible hysteresis in the system. Our results suggest that unstable and irreversible responses to warm anomalies are possible, and can lead to substantial changes in ice flux over relatively short periods of only a few decades.

## 1   Introduction

The Antarctic Ice Sheet has been losing mass in recent decades (Otosaka et al., 2023). Much of this mass loss has originated in the Amundsen Sea Embayment (ASE) sector in West Antarctica (Rignot et al., 2019), where glaciers have undergone rapid acceleration (Mouginot et al., 2014), inland thinning (Konrad et al., 2017), and widespread grounding-line retreat (Rignot et al., 2014). Between 1979 and 2017, ASE glaciers contributed over 7 mm to sea-level rise, which accounted for over half of the overall contribution from the entire Antarctic Ice Sheet (Rignot et al., 2019). This mass loss has led to concerns about the
current stability (Hill et al., 2023) and future evolution of the region (Cornford et al., 2015; Alevropoulos-Borrill et al., 2020; Reese et al., 2023).

The ASE glaciers are susceptible to the marine ice sheet instability (Weertman, 1974; Schoof, 2007), where ice flux increases as the grounding line retreats into deeper bedrock, giving rise to increased mass loss and further retreat. Under future climate conditions, an instability in this region has the potential to destabilize and collapse the sector (Feldmann and Levermann, 2015),
which contains 3.4 million km$^3$ of ice, or a sea level equivalent of over 5 m (Morlighem et al., 2020). Modelling studies have

predicted possible unstable episodes occurring in ASE in the future (Favier et al., 2014; Rosier et al., 2021; Reese et al., 2023), and recent results suggest that a phase of irreversible retreat has happened in the last 100 years (Reed et al., 2023).

The mass imbalance and accelerated ice flow in ASE has been attributed to reduced buttressing along coastal margins, where floating ice shelves have thinned due to ocean-driven melting (Gudmundsson et al., 2019; Pritchard et al., 2012; Paolo et al., 2015). Along the Amundsen Sea coastline, warm modified Circumpolar Deep Water (CDW) flows onto the continental shelf and into ice-shelf cavities (Jacobs and Hellmer, 1996; Dutrieux et al., 2014), causing some of the highest basal melt rates around Antarctica at an average of 14 to 27 m $yr^{-1}$ (Adusumilli et al., 2020). The influx of CDW, and therefore available heat content beneath the ice, varies on seasonal to decadal timescales (Dutrieux et al., 2014; Jenkins et al., 2016; Webber et al., 2017; Jenkins et al., 2018), and is strongly influenced by local wind forcing, natural climate variability and anthropogenic forcing (Thoma et al., 2008; Steig et al., 2012; Holland et al., 2022).

Changes in the thickness of the CDW layer impact the depth of the ocean thermocline and corresponding melt rates, and therefore affect the flow of ASE glaciers (Dutrieux et al., 2014; Jenkins et al., 2018; Gudmundsson et al., 2019). A shallow thermocline (600 m) in the mid to late 2000s coincided with widespread acceleration (Mouginot et al., 2014), enhanced thinning (Konrad et al., 2017) and grounding-line retreat (Rignot et al., 2014). Conversely, a deep thermocline (800 m) in 2012, following a strong La Niña event in 2011, caused low basal melt rates and possibly led to reduced glacier acceleration across the sector (Mouginot et al., 2014; Dutrieux et al., 2014). Although these shifts in oceanic forcing appear to have had simultaneous impacts across the region, glaciers have been shown to respond differently depending on their local geometry and bed topography (Scheuchl et al., 2016).

Between the late-1990s and mid-2000s, while most ASE glaciers experienced reduced acceleration, possibly in response to cooler ocean conditions (Mouginot et al., 2014; Dutrieux et al., 2014; Naughten et al., 2022), Pine Island Glacier (PIG) continued accelerating (Rignot et al., 2002; Mouginot et al., 2014) and thinning (Shepherd et al., 2001; Wingham et al., 2009). The glacier had been retreating across an ice plain since the early 1990s (Park et al., 2013; Corr et al., 2001), where its grounding line had been situated on the seaward side of a prominent seabed rise following an earlier slow down (Mouginot et al., 2014; Jenkins et al., 2010). Although the initial cause of this recent retreat is unknown, it is clear that the subsequent mass loss was unaffected by the reduced basal melt rate in the early 2000s (Dutrieux et al., 2014). This may be due to internal ice dynamics becoming the dominant driver of retreat after the initial warm perturbation (Reed et al., 2023).

Sediment cores recovered from beneath Pine Island ice shelf indicate a similar scenario occurring in the 1940s, when PIG was grounded on a large subglacial ridge, 40 km downstream of its present-day position (Smith et al., 2017). Following a climate anomaly in West Antarctica (Schneider and Steig, 2008), possibly caused by the 1939–42 El Niño event, a pre-existing cavity beneath the ice shelf became connected with the open ocean. In subsequent years, when conditions returned to pre-anomaly levels (Schneider and Steig, 2008), the cavity connection remained open and the grounding line continued to retreat down the landward, retrograde slope of the ridge. Whilst we have no observations of ocean conditions in the Amundsen Sea prior to 1994, modelling results suggest there was a similar variability that we observe today (Dutrieux et al., 2014; Naughten et al., 2022). Therefore, despite likely subsequent periods of colder forcing, the glacier has not recovered to its original position on the ridge. This suggests that the retreat had entered an unstable and irreversible phase after the 1940s climate anomaly, which

had finished when the glacier reached a shallower section of bed around 1990 (De Rydt and Gudmundsson, 2016; Reed et al., 2023; Mouginot et al., 2014; Park et al., 2013).

In this study we investigate the transient response of PIG to changes in basal melting when it is grounded on the subglacial ridge, 40 km downstream from its present-day position. We extend the work of Reed et al. (2023) to address the following questions: (1) Could a short period of increased basal melt initiate retreat from the ridge? (2) Once PIG is retreating, does a return to colder conditions with lower melt stop the retreat and allow for recovery back to the ridge? (3) What is the sensitivity of irreversibility to different basal melt rates? By answering these questions, we can assess how local geometry and external forcing may have impacted the retreat of PIG from the ridge in the 1940s. In Sect. 2, we give a description of the model and domain used in this study and explain the experiment setup. In Sect. 3 we present the results from each of the experiments and discuss the findings in Sect. 4.

## 2 Methods

For this study, we use the finite-element, vertically-integrated ice-flow model Úa (Gudmundsson et al., 2012) to solve the shallow ice-stream approximation (SSTREAM or SSA; see Appendix A) (e.g., Macayeal, 1989) for a regional configuration of PIG. Úa has been used in several studies from diagnostic (Reese et al., 2018; Mitcham et al., 2022; Sun and Gudmundsson, 2023) to transient (Hill et al., 2021; Jones et al., 2021) investigations, tipping point analysis (Rosier et al., 2021; Reed et al., 2023), ice-ocean coupled experiments (De Rydt and Gudmundsson, 2016; De Rydt and Naughten, 2023) and intercomparison projects (Pattyn et al., 2013; Cornford et al., 2020; Levermann et al., 2020).

### 2.1 Model domain and mesh

The model domain encompasses the entire grounded catchment of PIG (Mouginot and Rignot., 2017) and its floating ice shelf, a total of $\sim 188,000$ km$^2$ (Fig. B1). The calving front position, from BedMachine Antarctica, v2 (Morlighem et al., 2020), corresponds approximately to the 2008/09 front and remains fixed throughout the experiments. Aerial photographs show the 1940s calving front to be in a similar position (Rignot, 2002). Dirichlet boundary conditions are imposed on the grounded section of the boundary with a zero velocity along the ice divides, and Neumann boundary conditions are imposed along the ice front using ocean pressure.

For the model inversion (Sect. 2.3) and setup of the approximate 1940s geometry (Sect. 2.5), a fixed irregular triangular mesh was used that was generated with MESH2D (Engwirda, 2014). This consisted of 29,797 nodes and 58,777 linear elements and was refined on the ice-shelf ($\sim$0.5 to 1 km) and in areas of high strain rate and strain-rate gradients ($\sim$1 to 2 km). A coarser mesh size was used for regions of slow moving ice far from the main tributaries, with elements sizes of $\sim$10 km (Fig. B2). The minimum, median, and maximum element sizes for the fixed mesh were 563 m, 1,311 m and 11,330 m respectively.

For cold and warm transient experiments (Sect. 2.6) a further time-dependent mesh refinement was applied around the grounding line, adapting the mesh as the geometry evolved every half a year. This refinement ensured 500 m mesh elements

within 5000 m of the grounding line, and 250 m elements within 2000 m of the grounding line (Fig. B2). The grounding line in Úa is defined by the floatation condition and closely matches its present-day position (Mouginot and Rignot., 2017).

## 2.2 Input data

The aim of this study is to model the response of a 1940s PIG to changes in basal melting, but very little data are available prior to the satellite observation period, which began in the 1970s. To overcome this, we start from a present-day PIG configuration and let the ice stream evolve in time, without any basal melting applied, so it advances forward to the subglacial ridge downstream (Fig. 1). The present-day ice thickness, surface elevation, bedrock topography and ice density were supplied from BedMachine Antarctica, v2 (Morlighem et al., 2020). These geometry variables have a nominal date of 2015 and a resolution of 500 m, and each was linearly interpolated onto the model mesh. Some small adjustments were made to the ice-shelf thickness in the ice plain area to ensure the hydrostatic floating condition was met for the PIG ice shelf. These updated data were provided by Mathieu Morlighem and later incorporated into Bedmachine v3. The maximum change was a thickness decrease of 250 m, but generally there were decreases of between 80 m and 100 m. The upper surface accumulation was given by RACMO2.3p2 and averaged between 1979 to 2016 (Wessem et al., 2018).

## 2.3 Model inversion

The initial conditions for a present-day PIG configuration were generated using an inverse method. By making use of Úa's optimization capabilities, we estimated the unknown model parameters of basal slipperiness ($C$) and ice rate factor ($A$) using known velocity measurements from the MEaSUREs Annual Antarctic Ice Velocity Maps dataset (Mouginot et al., 2017a, b). Further details of the inversion process are given in Appendix C, with the final results shown in Fig. C1. There is a good fit between modelled and observed velocities, with only large differences where we are missing measurements near the ice front. The mean difference, excluding these large discrepancies, is 0.94 m yr$^{-1}$, with typical differences within 30 m yr$^{-1}$ in all the main tributaries.

## 2.4 Basal melt-rate parameterization

To simulate changes in ocean conditions in the perturbation experiments (Sect. 2.6), we use a depth-dependent melt-rate parameterization applied beneath the ice shelf (Fig. 2b). This allows us to directly link observations of the vertical stratification of ocean conditions to our vertical basal melt profile. The parameterization represents typical conditions in Pine Island Bay, which has a shallow cold layer, a deep warm layer and a rapidly changing ocean thermocline between them (Dutrieux et al., 2014). Similar to previous studies (Favier et al., 2014; De Rydt and Gudmundsson, 2016; Reed et al., 2023), the parameterization uses a piecewise-linear function of depth with zero melt in the shallow and 100 m yr$^{-1}$ in the deep (Bindschadler et al., 2011; Dutrieux et al., 2013; Shean et al., 2019), and these are separated by a 400 m thick thermocline.

In the cold experiments, the shallow zero melt layer extends down to 400 m depth and the deep layer begins at 800 m depth (Fig. 2b). We refer to this cold parameterization as having a thermocline depth of 800 m to keep consistent with previous studies

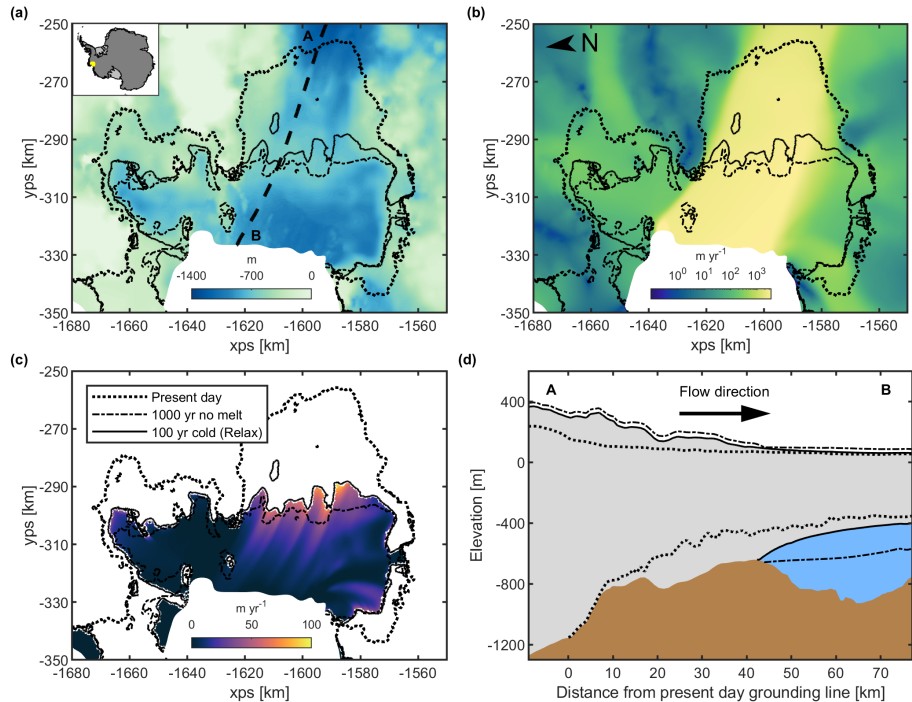

**Figure 1.** Bedrock elevation (**a**), ice surface speed (**b**) and basal melt rate (**c**) with overlain grounding lines for the initial model setup of the present-day configuration (black dotted), the steady-state geometry after 1000 years of no basal melting (black dash-dotted) and the steady-state result of the relaxation run after 100 years of cold forcing (black solid). Inset in **a** shows the location of PIG within Antarctica, using a polar stereographic projection (xps, yps). The speed in **b** and the melt rate in **c** are from the end of the relaxation simulation. The isolated grounding line upstream of the ridge in **a**, **b** and **c** indicates a cavity that forms where ice has thinned above deep bedrock. (**d**) Corresponding flowline profiles for the location shown as a thick dashed black line in **a**, with the flow direction from A to B.

(Favier et al., 2014; De Rydt and Gudmundsson, 2016; Reed et al., 2023). This forcing is based on the deepest thermocline and coldest conditions observed in Pine Island Bay between 2012 and 2013 (Dutrieux et al., 2014; Webber et al., 2017). In the

warm experiments, the thermocline is shifted upwards by 200 m, so has a depth of 600 m. This is representative of the warmest conditions and shallowest thermocline recorded in Pine Island Bay in 2009 (Dutrieux et al., 2014).

     Although this is a simplified parameterization, and does not capture feedbacks between ice, ocean and bed, it allows us to draw conclusions about the direct effect of basal melting, which varies on a decadal scale with the changing ocean conditions. The basal melt was applied to mesh elements that were strictly downstream of the grounding line, and fully floating, to ensure

we did not overestimate mass loss (Seroussi and Morlighem, 2018).

## 2.5    Advance to ridge and relaxation

To generate an approximate 1940s PIG configuration at the ridge we start from the present-day setup obtained in the inversion stage (Sect. 2.3), which is shown by the dotted line in Fig. 1. PIG has several tributaries flowing into the landward sides of

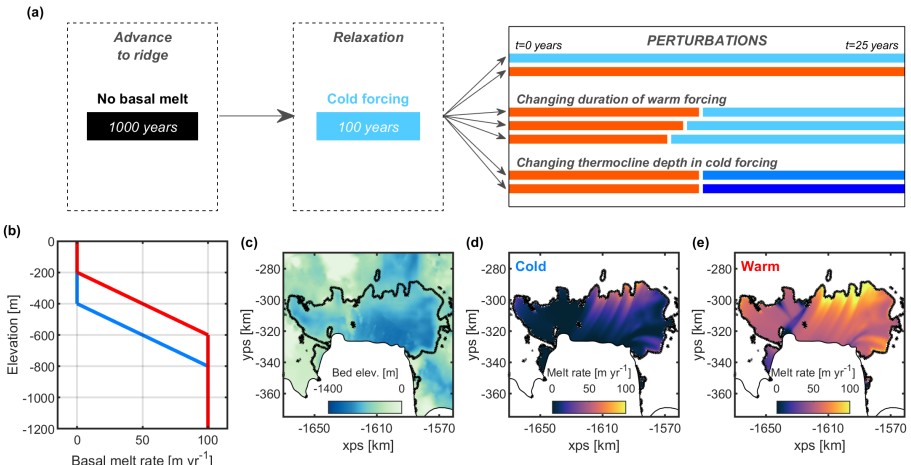

**Figure 2.** (**a**) Schematic of the experiment setup. For the initial phase, the model is run for 1000 years with no basal melting to allow for a quasi-steady state to be reached on the subglacial ridge. This is followed by 100 years of cold forcing to put the model into a new state using basal melting beneath the ice shelf. This represents an approximate configuration of PIG prior to the 1940s retreat from the ridge and provides the starting point for the perturbation experiments. Different periods of warm and cold forcing are then applied until 25 years is reached. (**b**) Depth-dependent melt-rate parameterization for the cold (blue) and warm (red) forcing. (**c**) Bed elevation, (**d**) basal melt rate for the cold parameterization and (**e**) basal melt rate for the warm parameterization, at the start of the perturbation experiments. In (**c**) – (**e**) the grounding line is shown as a thick black line and model boundary as a thin black line.

its ice shelf, with the main central trunk flowing the fastest in a north-west direction. The grounding line of the main trunk is
located at a depth of 1200 m on the bottom section of a retrograde slope, 45 km upstream from the subglacial ridge.

From the present-day configuration, we run the model with no basal melting to allow the ice stream to thicken and advance forward to the ridge. This is run for 1000 years to ensure a new quasi-steady state can be reached. Previous modelling results show that there is a steady-state position at the ridge when a deep thermocline (>1000 m) is used in the melt parameterization (Reed et al., 2023), which gives close to zero melt everywhere for this geometry. Hence, we use zero melt rather than the cold
conditions described in Sect. 2.4, as previous results in Reed et al. (2023) show that a 800 m deep thermocline would not be sufficient to advance from the present-day position to the ridge.

Within the first 25 years, the ice shelf sufficiently thickens so that it grounds on the landward side of the ridge and the grounding line advances 40 km. By the end of the simulation the ice stream only advances a further 5 to 10 km, resulting in a thick ice shelf grounded at the front of the ridge crest. Hence, the subglacial ridge provides a steady-state position for PIG,
which does not advance beyond it despite the absence of basal melting. This is also aided by the fixed calving front, which is not far from its 1940s position (Rignot, 2002; Arndt et al., 2018). It is unlikely that a slightly more advanced calving front would provide much additional buttressing, (Fürst et al., 2016), so would have a limited impact on subsequent ice dynamics.

After setting up the new steady state on the ridge we relax the ice geometry to get an approximate 1940s PIG configuration, with a more realistic ice shelf draft. This is done by running the model with the cold basal melt parameterization described

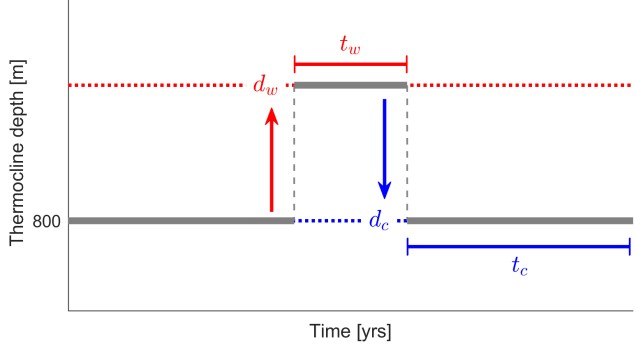

**Figure 3.** Schematic showing an example of the warm ('w', red) and cold ('c', blue) forcing durations ($t_w$, $t_c$) and thermocline depths ($d_w$, $d_c$) in the perturbation experiments.

in Sect. 2.4; this has a thermocline depth of 800 m, and therefore the maximum melt is deeper than the crest of the ridge. However, due to the thick ice shelf at the start of the transient simulation, this initially causes high melt rates, with a mean of 40 m yr$^{-1}$ and integrated melt of 97 Gt yr$^{-1}$. We run the model for 100 years which is enough time to allow the ice stream to adjust to the updated forcing and reach a new quasi-steady state with basal melting applied (Fig. 1).

Within the first five years the melt rate decreases by almost 50 %, and after 15 years the mean melt rate has dropped below 18 m yr$^{-1}$ and the integrated melt is below 40 Gt yr$^{-1}$, remaining at approximately this level until the end of the simulation. This level of melting is typical for cold years in Pine Island Bay (Dutrieux et al., 2014). At the end of the simulation, both grounded and floating ice have thinned and the ice stream has sped up by 6 to 8 %. This causes a 2 to 4 km retreat along the main trunk from the shallowest section of ridge, and up to 10 km along the deeper southern channel. However, the ice stream remains grounded along the top of the ridge, as shown in Fig. 1a, with no connection being established between the outer open ocean and the inner region upstream of the ridge. This new state represents the approximate situation prior to the warm anomaly in the 1940s (Smith et al., 2017) and is the starting configuration for the perturbation experiments (Sect. 2.6).

### 2.6 Perturbation experiments

Following the model setup and relaxation, we ran a suite of perturbation experiments with different warm and cold forcing. The melt rate below the thermocline is kept constant at 100 m yr$^{-1}$ in all experiments, but the depth of the thermocline is varied to make direct comparisons with ocean observations (Dutrieux et al., 2014; Jenkins et al., 2018) and to be consistent with previous studies (De Rydt et al., 2014; De Rydt and Gudmundsson, 2016; Bradley et al., 2022). A summary of these experiments is given in Table 1 and a schematic of the entire model setup is shown in Fig. 2a. An example of the perturbation experiments is given in Fig. 3. These experiments were designed to assess the sensitivity of a 1940s PIG to changes in basal melt, over a 25 year period, by varying the forcing history on a near-decadal timescale. This will help us to better understand the reversibility of retreat from the subglacial ridge in the 1940s.

**Table 1.** A summary of the forcing durations and thermocline depths for the basal melt-rate parameterization in the perturbation and sensitivity experiments (Sect. 2.6). Bold numbers indicate the parameter being changed. The first eight experiments are individual simulations and are illustrated in the schematic in Fig. 2. Note that WARM12 and COLD800 are the same test but are shown twice for comparison purposes. The final two experiments (WARMvar and COLDvar) test a range of parameter values.

| Experiment | Warm duration $(t_w)$ [yrs] | Warm thermocline depth $(d_w)$ [m] | Cold duration $(t_c)$ [yrs] | Cold thermocline depth $(d_c)$ [m] |
|---|---|---|---|---|
| COLD25 | 0 | n/a | 25 | 800 |
| WARM25 | 25 | 600 | 0 | n/a |
| WARM10 | **10** | 600 | 15 | 800 |
| WARM11 | **11** | 600 | 14 | 800 |
| WARM12 | **12** | 600 | 13 | 800 |
| COLD800 | 12 | 600 | 13 | **800** |
| COLD1000 | 12 | 600 | 13 | **1000** |
| COLD1200 | 12 | 600 | 13 | **1200** |
| WARMvar | 2, 3, 4, 5, 6, 10 | **400** | 50 | 800 |
| | 2, 3, 4, 5, 6, 10 | **450** | 50 | 800 |
| | 2, 4, 5, 6, 7, 8, 9, 10 | **500** | 50 | 800 |
| | 5, 6, 7, 8, 9, 10 | **550** | 50 | 800 |
| | 5, 9, 10, 11, 12, 15 | **600** | 50 | 800 |
| | 15, 16, 17, 18, 19, 20 | **650** | 50 | 800 |
| | 35, 40, 41, 42, 43, 44, 45, 50 | **700** | 50 | 800 |
| COLDvar | 12 | 600 | 100 | **800** |
| | 12 | 600 | 100 | **900** |
| | 12 | 600 | 100 | **1000** |
| | 12 | 600 | 100 | **1100** |
| | 12 | 600 | 100 | **1200** |

Before investigating the reversibility, we first ran two experiments with fixed forcing for 25 years (COLD25 and WARM25). We used these simulations as control cases to compare the time varying perturbation experiments against. Whilst we do not expect these extended periods of forcing to be realistic, they act as lower and upper estimates of the dynamical response of PIG.

The second set of experiments compared the impact of different durations of warm forcing on the retreat of PIG from the ridge (WARM10–12). The aim of these is to determine whether short warm anomalies are sufficient to force PIG off the ridge

and initiate a retreat. We applied cold forcing after each of the warm anomalies, until 25 years, to assess the reversibility of retreat.

The third set of perturbation experiments tested the impact of different cold forcing that follows the warm anomaly (COLD800–1200). The aim of these simulations is to explore whether typical cold conditions are sufficient to stop an already retreating PIG and allow it to recover back to the ridge, or whether more extreme forcing is needed. To adjust the cold conditions, we lowered the depth of the thermocline in the melt-rate parameterization to 1000 m and 1200 m. The colder forcing was applied once PIG had already started retreating and melting had begun upstream of the ridge. This is to replicate the known situation in the 1940s (Schneider and Steig, 2008; Smith et al., 2017).

The final set of experiments test the sensitivity of irreversible retreat for a wider suite of forcing conditions (WARMvar and COLDvar). All model simulations start at the ridge and consist of a period of warm forcing, followed by cold forcing. This allowed us to test whether any retreat was irreversible or not. We first experimented with the warm anomaly, by changing the duration of forcing (between 0 and 50 years) and the thermocline depth (400 to 700 m), where each of the warm perturbations was followed by a 50 year period of cold forcing with an 800 m thermocline depth. The warm forcing here spans the shallowest thermocline depths observed in Pine Island Bay (Dutrieux et al., 2014; Webber et al., 2017) and predicted under future conditions (Naughten et al., 2023). In total, there were 46 WARMvar model simulations with varying durations of warm forcing and thermocline depths. Not all combinations of parameters were tested as we were only interested in when the irreversible transition occurred.

The next experiment varied the cold forcing, after an initial warm anomaly, by changing the thermocline depth (800 to 1200 m) and then finding which simulation had a reversible retreat. These five simulations all ran for 100 years, and had the same initial warm forcing of a 600 m thermocline for 12 years, so that melting had already started upstream of the ridge. Although the deepest thermocline observed in Pine Island Bay was 800 m in 2012 to 2013, we include deeper thermoclines to account for possible cold convection events occurring earlier in the twentieth century (Naughten et al., 2022).

## 3 Results

### 3.1 Constant forcing

The first experiment, COLD25, reveals that 25 years of cold forcing, after the initial relaxation run, makes no impact on the ice thickness, speed or grounding-line position of PIG. The glacier remains in a steady state, with a balance between the mass gained over the entire domain and mass lost through the calving front and to basal melting. Throughout the experiment, the mean melt rate remained at approximately 14 m yr$^{-1}$ and the integrated melt was 33 Gt yr$^{-1}$.

In contrast to the constant cold run, when warm forcing was applied for 25 years, in the WARM25 experiment, the ice shelf thinned by 100 to 400 m and sped up by 1000 m yr$^{-1}$ (Fig. 4). Due to a loss of buttressing, ice upstream of the grounding line thinned by up to 100 m and the grounding line retreated 20 to 30 km. Over the 25 years, as the ice shelf thinned, mean melt rates decreased by 50%, from 57 m yr$^{-1}$ at the start of the run down to 25 m yr$^{-1}$ at the end. Integrated melt was initially high at 158 Gt yr$^{-1}$, but this rapidly decreases to a minimum of 84 Gt yr$^{-1}$ after 10 years (Fig. 5e). There is an increase in

integrated melt as the upstream cavity opens after 12 years but as the glacier reaches its final position the melt decreases again to 93 Gt yr$^{-1}$. The melt values from five years onwards are typical for a warm year in Pine Island Bay (Dutrieux et al., 2014).

In the first 5 to 10 years, there is little retreat across the top of the ridge, but there is an inland propagation of thinning and acceleration along the main trunk (Fig. 4). This leads to an increase in ice flux across the grounding line (Fig. 5c), and further thinning over a depression in the bed causes isolated cavities to form upstream of the grounding line (Fig. 4). After two more

215  years of melting, the small cavities merge with each other and then connect with the main outer cavity via the southern channel, causing basal melting upstream of the ridge.

Once high basal melting occurs upstream of the ridge, and is sustained, the grounding line continues to retreat down the retrograde sloping bed (Fig. 4). An ice rumple begins to form after 12 years and ice flux rapidly increases (Fig. 5c) as the ice thins and retreats across the deeper parts of the bed. The ice rumple becomes more isolated and reduces in size, leaving only a

small region of the ice shelf still grounded on the ridge. This is the sequence of events that happened in the 1970s (Jenkins et al., 2010; Mouginot et al., 2014). Eventually, after 18 years, the ice shelf completely detaches from the ridge and the grounding line stabilizes on the prograde slope of a seabed rise 20 to 30 km upstream of the ridge (Fig. D1). The glacier stops accelerating and there is a gradual decrease in ice flux across the grounding line as the glacier approaches a new state (Fig. 5c). This is consistent with observations of PIG stabilizing at an ice plain in the early 1990s (Park et al., 2013; Mouginot et al., 2014).

## 3.2  Duration of warm forcing

The constant forcing experiment showed that sustained warming for 25 years leads to substantial thinning and acceleration, causing a 20 to 30 km retreat. Our second set of experiments tests whether shorter, more realistic, warm anomalies would be sufficient to force PIG off the ridge and initiate a retreat. To best represent the 1940s situation, we first applied warming for 12 years to ensure melting occurs upstream of the ridge. This then represents the oceanic connection being established between

the inner and outer cavities (Smith et al., 2017). After the warming, we subsequently applied cold forcing until 25 years to represent the shift back to normal conditions after the 1940s El Niño event (Schneider and Steig, 2008).

Figure 6 shows that despite stopping the warm anomaly after 12 years, the grounding line continues to retreat to the ice plain upstream, without any additional forcing being applied. Following the warm anomaly, when cold forcing is re-applied, the mean melt rate immediately decreases from 31 m yr$^{-1}$ to 11 m yr$^{-1}$ and the integrated melt decreases from 84 Gt yr$^{-1}$ to

19 Gt yr$^{-1}$, which is lower than at the end of the constant cold simulation (Fig. 5e). This is because the ice shelf is thinner than when it is grounded on the ridge. Despite this drop in overall melt, it is insufficient to stop the increase in ice flux or reverse the grounding line retreat.

We also ran shorter warm anomalies of 10 and 11 year durations, where a switch to cold forcing occurs before the melting starts upstream of the ridge. Figure 5a,c,e shows that after 11 years of warming, there is a continued retreat with a loss of

grounded area and increased ice flux, and the final configuration coinciding with the 12 year simulation. When warm forcing was stopped after 10 years, at least a year before there is melting upstream, this allowed the ice shelf to thicken, leading to an increase in buttressing. Consequently, upstream thinning was reduced, and there was no connection between the inner

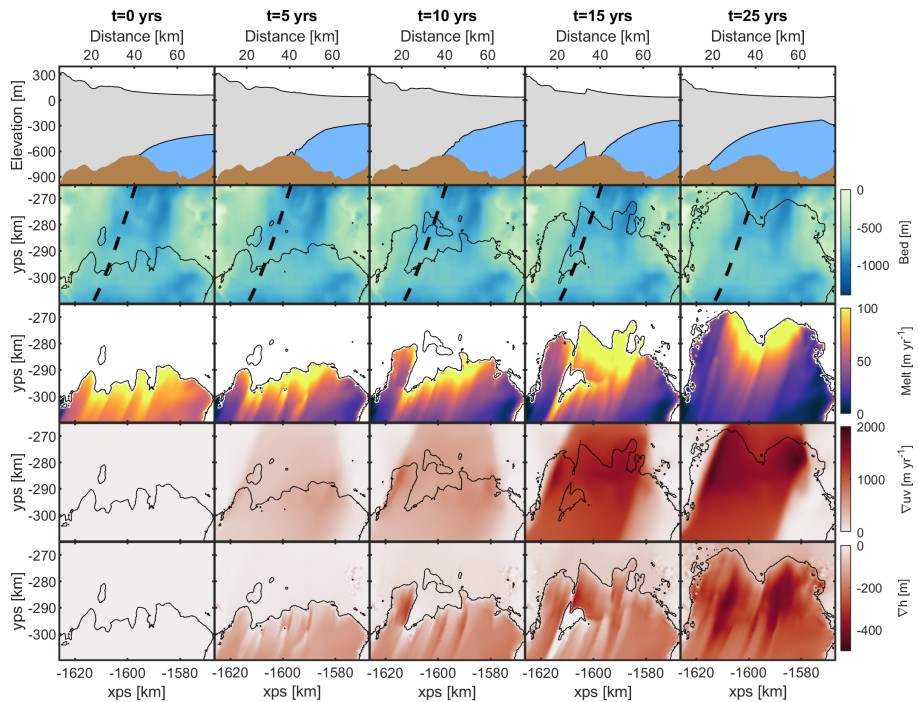

**Figure 4.** Evolution of flowline profiles (**top row**), grounding-line position over bed elevation (**second row**), basal melt rate (**third row**), change in ice speed (**fourth row**) and change in ice thickness (**bottom row**) during the WARM25 experiment. The columns from left to right show results after: $t = 0, 5, 10, 15, 25$ years. Changes in speed and thickness are with respect to $t = 0$ years.

upstream cavities and the main outer cavity. There was a decrease in ice flux and a re-advance of the grounding line, eventually recovering its original position on the ridge.

### 3.3 Magnitude of cold forcing

The third set of perturbation experiments show that only extreme cold forcing, which causes much lower melt, can stop and reverse the retreat of PIG from the ridge (Fig. 5b,d,f). After a warm anomaly for 12 years, we lower the thermocline in the melt parameterization to 1200 m, so the bottom of the upper cold layer is at 800 m, and the highest melt is below the depth of the cavity grounding line. This causes the overall melt to decrease to almost zero for the rest of the simulation. This was enough to stop the mass loss and decline in grounded area, and reverse the retreat. A more realistic cold parameterization, with a thermocline depth of 1000 m, gives mean melt rates of 2 to 5 m yr$^{-1}$ and integrated melt of 4 to 12 Gt yr$^{-1}$, which is a third of the original cold forcing. However, this is not sufficient to stop the already retreating grounding line.

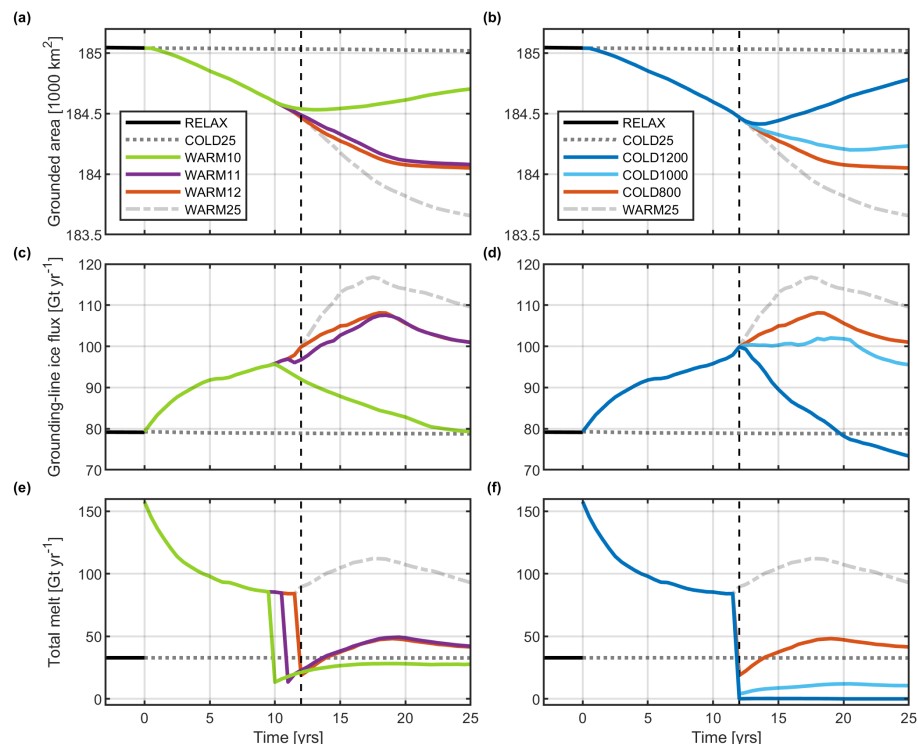

**Figure 5.** Time evolution of the total grounded area (**a,b**), ice flux across the grounding line (**c,d**) and integrated basal melt rate (**e,f**) for the reversibility experiments. The final three years of the relaxation run using cold forcing is shown in solid black, and the two control runs with continuous cold and warm forcing for 25 years are shown in dark grey dotted and light grey dash-dotted, respectively. The green, purple and red solid lines in **a**, **c**, and **e** show warming for 10, 11 and 12 years respectively, with cold forcing thereafter. The dark blue, light blue and red solid lines in **b**, **d**, and **f** show warming for 12 years followed by cold forcing using thermocline depths of 1200 m, 1000 m and 800 m respectively. The vertical black dashed line indicates the time of melting starting upstream of the ridge in the WARM25 experiment (12 yrs). Note, the WARM12 and COLD800 show the same result in both sets of panels. Details of these perturbations experiments are summarized in Table 1.

## 3.4  Mapping the stability regime

The previous experiments showed that the response of a 1940s PIG to changes in basal melting varies according to the duration
of warm forcing and the magnitude of cold forcing that follows. To expand these results, further model simulations were run to cover a wider test space of warm forcing parameters, followed by at least 50 years of cold forcing. Figure 7 and Fig. E1a illustrate the results of numerous simulations with different combinations of thermocline depths and durations of warm forcing. They show that a short warm period of only five to six years, with thermocline depths between 400 and 500 m, can cause irreversible retreat back to the ice plain, despite over 90 years of cold forcing after the warm anomaly. In contrast, melting

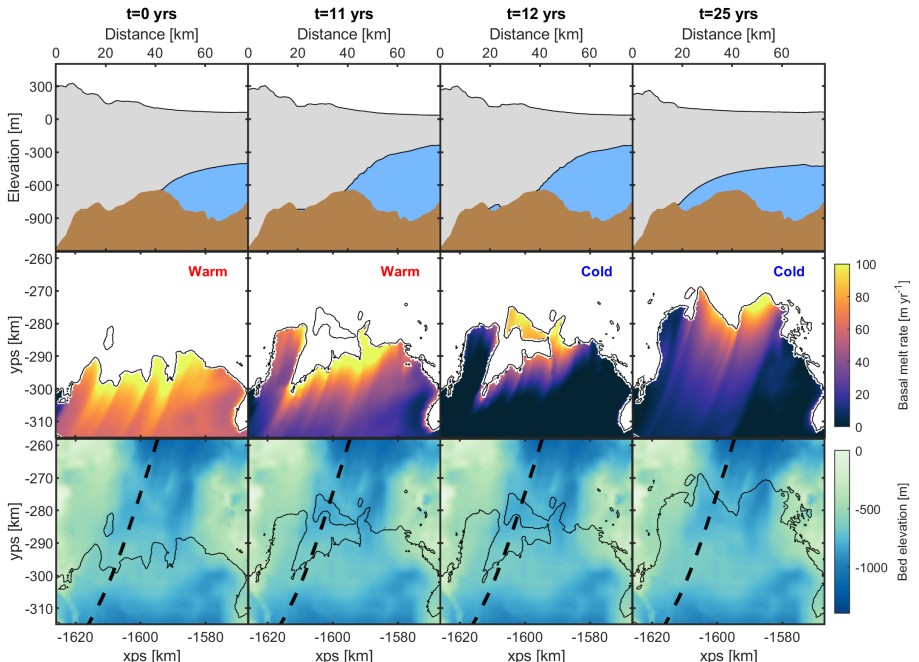

**Figure 6.** Evolution of flowline profiles (**top row**), basal melt rate (**middle row**) and grounding-line position on bedrock elevation (**bottom row**) for the WARM12 experiment. Warm forcing is applied for 12 years, until melting occurs upstream of the ridge, then cold forcing is applied thereafter to test for reversibility.

caused by a deeper thermocline of at least 700 m, during the warm period, has to be applied for at least 40 years to cause an unstable response.

To expand the cold forcing experiment, the previous simulations from Sect. 3.3 were run for longer to determine whether the initial grounding line retreat could eventually be reversed (Fig. E1b). However, as shown before, only unrealistic cold forcing with thermocline depths between 1100 and 1200 m, for at least 20 years, is sufficient to stop the retreat and allow a recovery back to the ridge. Whereas, more realistic cold forcing with thermocline depths shallower than 1000 m is insufficient, despite more than 80 years of forcing after the warm anomaly.

## 4 Discussion

Our modelling results show that the subglacial ridge beneath present-day PIG ice shelf provides a steady and stable grounding-line position after the glacier advances forward with no basal melting (Fig. 1). The glacier remains grounded on the ridge when forced with cold ocean conditions, despite some thinning of grounded and floating ice, and finds a new steady state on small prograde or flat sections along the ridge crest (Fig. D1). It remains in this position because the highest basal melt rates, which correspond to the warm Circumpolar Deep Water, are limited to below the depth of the ridge crest, resulting in little thinning

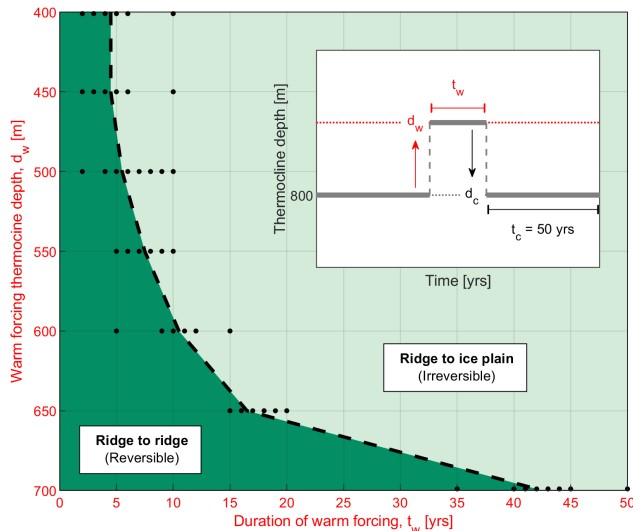

**Figure 7.** Reversibility of PIG depending on the duration of warm forcing ($t_w$) and the thermocline depth ($d_w$). The green areas illustrate two different final grounding-line positions (ridge and ice plain) after a period of warm forcing followed by cold forcing, with a starting position at the ridge. Dark green indicates parameters that lead to a final position at the ridge (reversible), and light green encompass all parameter combinations which lead to irreversible retreat back to the ice plain. Warm forcing is varied, then cold forcing is applied for 50 years with a 800 m deep thermocline. The dashed line between the two areas indicates the approximate change in behaviour between reversible and irreversible simulations, where we have used a discrete number of model simulations (markers) to generate this parameter space. Inset is a schematic of the model forcing setup.

(Fig. 2). This modelling result is consistent with sediment core observations that suggest PIG had been grounded on the ridge since the early Holocene (Smith et al., 2017).

## 4.1  Warm forced retreat

The first of our aims was to determine if a warm anomaly could initiate a PIG retreat from its stable position on the ridge, which is likely what happened in the 1940s (Schneider and Steig, 2008; Smith et al., 2017). To replicate warmer ocean conditions we applied a step change in forcing, which corresponded to a shallower thermocline and therefore higher basal melt rates (Fig. 2). During the first 10 years of warming there was thinning and acceleration (Fig. 4), which propagated far inland along the main trunk, leading to an increase in ice discharge across the grounding line (Fig. 5). Ice-shelf thinning is known to have upstream effects, through a loss of buttressing, especially where thinning is concentrated near to the grounding line or margins (Fürst et al., 2016; Reese et al., 2018). Previous modelling studies also show that thinning and acceleration can propagate upstream of the grounding line when a present-day PIG is forced by warm conditions (Favier et al., 2014; Seroussi et al., 2014). This has been observed for PIG and other neighbouring glaciers in the mid to late 2000s when thinning, acceleration and mass loss followed a period of warm ocean conditions in the Amundsen Sea (Dutrieux et al., 2014; Mouginot et al., 2014; Konrad et al.,

2017; Jenkins et al., 2018). Our modelling results combined with observations demonstrate that glacier flow and mass loss can be sensitive to changes in ocean conditions, when grounded on a topographic high, and we would expect other glaciers to respond similarly if they are in a comparable configuration.

The thinning and acceleration of grounded ice in our warm experiment led to the formation and merging of subglacial cavities upstream of the ridge (Fig. 4). These cavities occur where ice becomes marginally afloat over the deep lying bedrock, and this has also been shown in a previous idealized modelling study (De Rydt and Gudmundsson, 2016). If warm forcing continues until 12 years, the cavities connect with the main outer cavity on the south side of the shelf, where there is faster grounding-line retreat down a steep retrograde bed (Fig. D1). This connection between the inner and outer cavity leads to high basal melting upstream of the ridge. We can interpret this as the opening of the inner cavity to warm ocean waters (Smith et al.,

2017), which causes thinning and grounding-line retreat in the central trunk. When this happens, an ice rumple forms on the shallower bed and the glacier is no longer in a stable state.

Due to the limited number of observations of grounding line position (Jenkins et al., 2010; Smith et al., 2017; Park et al., 2013), we do not know the exact retreat history of PIG. However, these observations suggest that it took approximately 30 years between the inner cavity opening to ocean waters in the 1940s and the ice shelf detaching from the ridge in the late 1970s.

In our simulations this happens on a shorter timescale, of approximately 10 years. This could be due to the simplified melt forcing that we use which does not consider any geometric or topographic feedbacks that have been shown to delay retreat by 10 years (De Rydt and Gudmundsson, 2016). Furthermore, we are using approximate bed conditions and ice rheology inferred from present-day velocities, so we would also expect these parameters to impact the timescale of retreat.

## 4.2  Unstable response

The second aim of this study was to determine whether the retreat of PIG from the ridge could be stopped by removing the warm anomaly and returning to normal or cold basal conditions. This was motivated by the variable ocean conditions that are observed in the Amundsen Sea, which directly influence basal melt rates inside the ice-shelf cavity (Dutrieux et al., 2014). Our reversibility experiments show that removing the warm forcing after 12 years, or when high basal melting is already occurring upstream of the ridge, does not stop the retreating grounding line or ongoing mass loss (Fig. 6). After a connection is

established between the inner and outer cavities, there is a sharp increase in grounded and floating ice velocities and greater ice flow across the grounding line. This is likely due to a loss of buttressing as the ice shelf continues to thin and unground from the ridge (Gudmundsson et al., 2019). With a greater drawdown of upstream ice, there is further thinning and retreat, signalling a change in behaviour and the possible crossing of a stability threshold or tipping point (De Rydt and Gudmundsson, 2016; Reed et al., 2023).

The increase in grounding-line ice flux as the ice thins and retreats across the deep sections of bed agrees with observations of PIG ice shelf becoming detached from the ridge between 1975 and 1982 (Jenkins et al., 2010; Mouginot et al., 2014). A surface impression on the ice shelf disappears as contact is lost with the bed, consistent with observed behaviour in the 1970s (Jenkins et al., 2010). Following the ungrounding, our results show that the glacier stabilizes on the prograde slope of a seabed rise 20 to 30 km upstream of the ridge and there is a gradual decrease in ice flow. This is likely what occurred in the late 1980s

to early 1990s, when PIG was grounded in a similar position at an ice plain (Corr et al., 2001; Park et al., 2013). This period may also have coincided with cold ocean conditions (Thoma et al., 2008; Dutrieux et al., 2014; Jenkins et al., 2016), which would have facilitated the stabilization, although previous modelling results show that cold conditions were not necessarily needed for a steady state to occur there (Reed et al., 2023).

These results support the hypothesis that PIG underwent an unstable retreat from the subglacial ridge in the 1940s (Smith et al., 2017; Reed et al., 2023), which could have been initiated by a temporary increase in basal melt. The change in melt was possibly due to a shoaling thermocline, following a tropically forced climate anomaly in West Antarctica (Schneider and Steig, 2008; Jenkins et al., 2016). The irreversible retreat would have been unaffected by a reverse of ocean conditions in the following years and PIG continued losing mass through the 1970s and 1980s (Jenkins et al., 2010; Mouginot et al., 2014). Eventually, the ice shelf detaches from the ridge and the grounding line retreats to an upstream ice plain, which leads to a reduction in ice flux across the grounding line (Mouginot et al., 2014). This sequence of events demonstrates that although an increase in basal melt is the initial cause of mass imbalance and retreat, it can be the dynamical response that becomes the dominant driver of mass loss once the forcing is removed. The glacier only stops retreating when it reaches a shallow section of bed upstream. A similar result was shown by Favier et al. (2014) and Seroussi et al. (2014), where a temporary increase in basal melting beneath a present-day PIG can cause acceleration and irreversible retreat, despite returning to previous conditions.

## 4.3 Sensitivity of irreversibility

The final aim of this study was to investigate the sensitivity of irreversible retreat to different basal melt forcing. We know from observations that ocean conditions in the Amundsen Sea vary on interannual to decadal timescales (Dutrieux et al., 2014; Jenkins et al., 2016, 2018), which is partly influenced by the varying strength of westerly winds over the continental shelf break (Thoma et al., 2008; Steig et al., 2012). Future predictions under a high emissions scenario show persistent shelf break mean westerlies by 2100, which suggests there could be more prevalent warm ocean conditions in the Amundsen Sea (Holland et al., 2019). Therefore, these sensitivity experiments were designed as hypothetical scenarios to gauge the influence of more extreme ocean conditions on the retreat of a glacier from a prominent seabed ridge.

Our modelling results show that a 12 year period of typical present-day warm conditions, which uses a basal melt parameterization with a 600 m deep thermocline, is sufficient to cause irreversible retreat from the ridge (Fig. 7a). This same basal melt parameterization applied for 10 years or less can still lead to the formation of upstream cavities. However, they do not merge with the main outer cavity and thus the glacier is able to thicken and re-advance. The shallowest thermocline results, of 500 m and above, show that if there is a thicker layer of CDW on the shelf and therefore greater melting, as has been shown is possible in future projections (Naughten et al., 2023), it means that an unstable response of a glacier like this could be triggered in just five to six years. This suggests that a future phase of irreversible retreat could be initiated with just a short period of increased melting, and it is therefore not necessary for there to be either a sustained period of melting or an additional event after the initial anomaly has finished. Model simulations suggest that shelf temperatures and basal melting have increased over the past century, which if continues, could lead to this increased melting beneath the ice shelf (Naughten et al., 2022).

We also show that once PIG is retreating from the ridge after a warm anomaly, only extreme cold conditions, where melt decreases to zero very quickly, can stop the retreat (Figs. 5b,d,f and 7b). This indicates that a hysteretic behaviour exists in response to varying basal melt, where much lower melt is required to advance the glacier back to its original position (Favier et al., 2014; Reed et al., 2023). Lower melt is possible if the ridge blocks the deepest warm waters (Dutrieux et al., 2014; De Rydt et al., 2014), but the melt parameterization that is used here does not take that into account. However, in support of the results here, a coupled modelling study has shown that a switch to pre-anomaly conditions is not necessarily sufficient to stop a retreat from the ridge, despite blocking at the ridge (De Rydt and Gudmundsson, 2016).

The timescales of irreversible retreat that we show with these results should be treated with caution and not used as an exact prediction for future retreat. We have used a depth-dependent melt-rate parameterization which only captures the first order response to melting, and neglects any ice-ocean feedbacks or topographic influences. Furthermore, we only vary the thermocline depth and keep the melt rate constant in the deep. However, a previous study using the same parameterization showed that doubling the deep melt rate has a smaller impact compared to raising the thermocline (Favier et al., 2014). A tapering down of the melt rate at the deep grounding lines also has a limited impact on a retreating glacier that has a similar geometry to PIG (De Rydt and Gudmundsson, 2016).

More realistic simulations using a coupled ice-ocean model would further improve these results, and allow us to identify the additional influence that bed topography and cavity geometry has on the ocean forcing, which has been highlighted in previous studies (De Rydt and Naughten, 2023; Bett et al., 2024). We have also used present-day input data, such as ice velocities and geometry, to generate spatially varying slipperiness and ice rate factor fields that are unlikely to model the exact evolution of a 1940s PIG (De Rydt et al., 2021). However, this study clearly demonstrates the melt sensitivity of an approximate 1940s glacier as it retreats from the ridge. From our results we cannot conclude whether the unstable retreat from the ridge was caused by natural variability alone or a combination of factors (O'Connor et al., 2023), but do know that once the retreat started, it would have needed a large decrease in basal melting to overcome the ice dynamical response, and this may not have been possible because of anthropogenic change (Holland et al., 2022). Whilst PIG may now be in a stable state with respect to changes in external forcing (Hill et al., 2023), these modelling results give us an insight into the dynamical behaviour of a glacier that is forced with short periods of changing basal melt when grounded on a topographic high.

## 5  Conclusion

This study set out to map the stability regime of PIG with respect to ocean-induced melt as it detached from a subglacial ridge 40 km downstream from its present-day grounding-line position. Our ice sheet modelling results show that the ridge provides a stable position for the glacier when there is either low or zero basal melting due to cold ocean conditions. We found that an increase in basal melt rates, which are observed during typical warm years in the Amundsen Sea, can cause substantial thinning and acceleration and are sufficient to force the glacier off the ridge, initiating a retreat. After 12 years of warm forcing, once high melting occurs upstream of the ridge, the retreat becomes irreversible despite a removal of the warm anomaly and a return to cooler conditions. Much lower melt is required to stop the retreat and allow a recovery back to the ridge, which

indicates there is a hysteresis in response to basal melting. Melt sensitivity experiments show that shorter, sub-decadal, warm anomalies with higher melt rates can also lead to irreversible retreat. We have used a simplified depth-dependent melt-rate for these experiments, but this has shown the direct effect of variable basal melt on the retreat of PIG from the ridge. Future modelling experiments that use more realistic ocean forcing, taking into consideration the effects of topography and cavity geometry, will increase certainty of the effects of changing melt on PIG and the wider West Antarctic Ice Sheet. However, it is unlikely that our conclusions will change owing to the hysteresis being a robust feature in our modelling experiments which incorporate a wide range of ocean forcing. The hysteresis appears primarily related to the broader geometrical features of the bed, in particular the existence of a prominent subglacial ridge, as well the general geometrical setting of PIG allowing for the formation of a confined ice shelf. These results demonstrate that if warm conditions, and higher basal melt, become more prevalent in the Amundsen Sea, an unstable response of glaciers in a similar configuration cannot be ruled out.

## Appendix A: Shallow ice-stream approximation

The shallow ice-stream equations in compact form are

$$\nabla_{xy} \cdot (h\boldsymbol{R}) - \boldsymbol{t}_{bh} = \rho_i gh \nabla_{xy} s + \frac{1}{2} gh^2 \nabla_{xy} \rho_i, \tag{A1}$$

where $\boldsymbol{R}$ is the resistive stress tensor defined as

$$\boldsymbol{R} = \begin{pmatrix} 2\tau_{xx} + \tau_{yy} & \tau_{xy} \\ \tau_{xy} & 2\tau_{yy} + \tau_{xx} \end{pmatrix}, \tag{A2}$$

with $\tau_{ij}$ the components of the deviatoric stress tensor, and

$$\nabla_{xy} = (\partial_x, \partial_y)^T. \tag{A3}$$

In Eq. (A1), $s$ is the ice upper surface elevation, $h$ is the ice thickness, $\boldsymbol{t}_{bh}$ is the horizontal component of the bed-tangential basal traction $\boldsymbol{t}_b$, $g$ is gravitational acceleration and $\rho_i$ is the vertically averaged ice density.

The deviatoric stresses $\tau_{ij}$ are related to the strain-rates $\dot{\epsilon}_{ij}$ using Glen's flow law

$$\dot{\epsilon}_{ij} = A\tau^{n-1}\tau_{ij}, \tag{A4}$$

where $\tau$ is the second invariant of the deviatoric stress tensor

$$\tau = \sqrt{\tau_{ij}\tau_{ij}/2}, \tag{A5}$$

$A$ is a spatially varying ice rate factor determined using inverse methods (Sect. 2.3) and $n = 3$ is a creep exponent.

For grounded ice, the basal traction is given by a Weertman sliding law

$$\boldsymbol{t}_b = C^{-1/m}||\boldsymbol{u}_b||^{1/m-1}\boldsymbol{u}_b, \tag{A6}$$

where $\boldsymbol{u}_b$ is the horizontal component of the bed-tangential ice-velocity, $C$ is a spatially varying slipperiness parameter determined using inverse methods (Sect. 2.3) and $m = 3$, which gives a non-linear viscous relationship.

## Appendix B: Mesh domain and element sizes

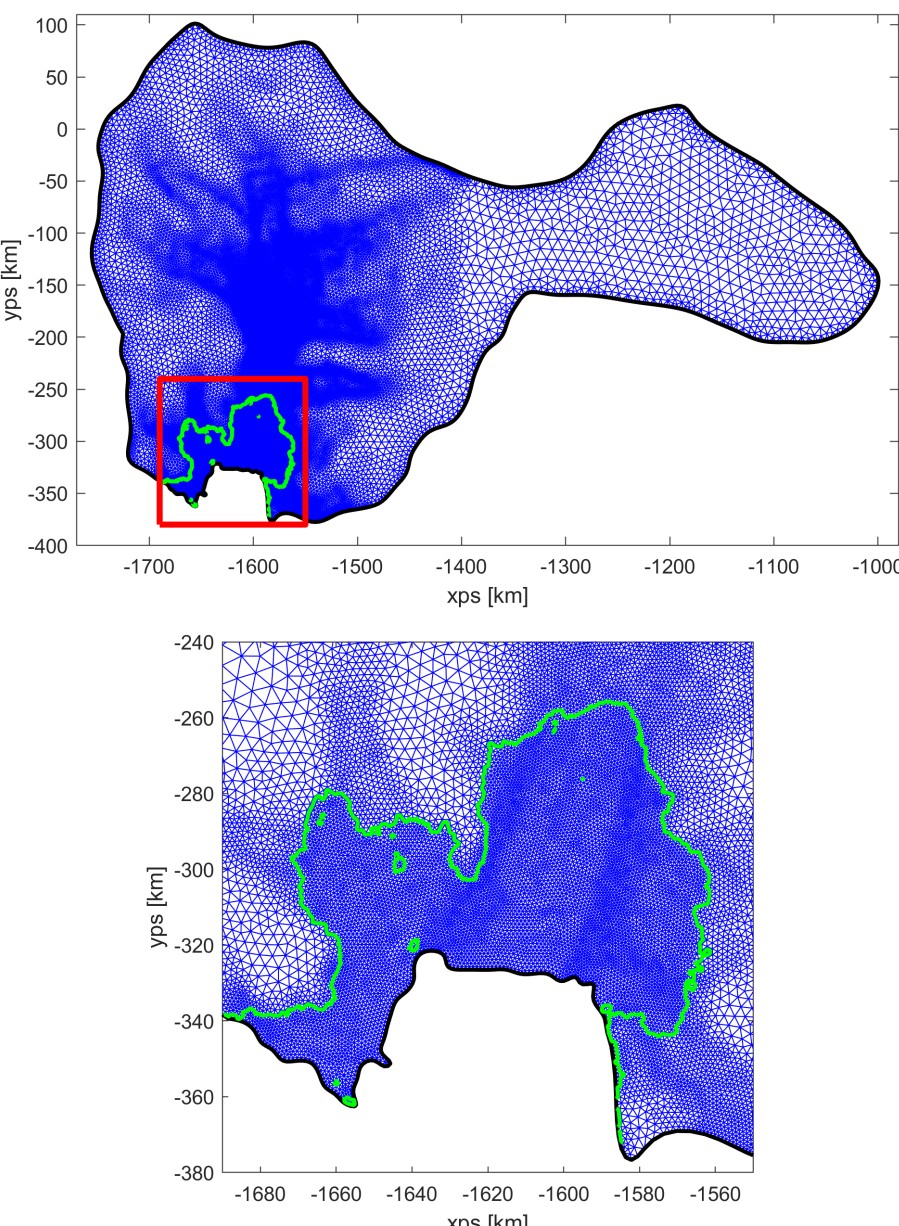

**Figure B1.** Initial mesh for the Pine Island Glacier model domain. Catchment area is encompassed by the thick black line and the linear triangular mesh elements are shown in blue. The red square in the top plot outlines Pine Island Ice Shelf which is enlarged in the lower plot, and both show the grounding line in green.

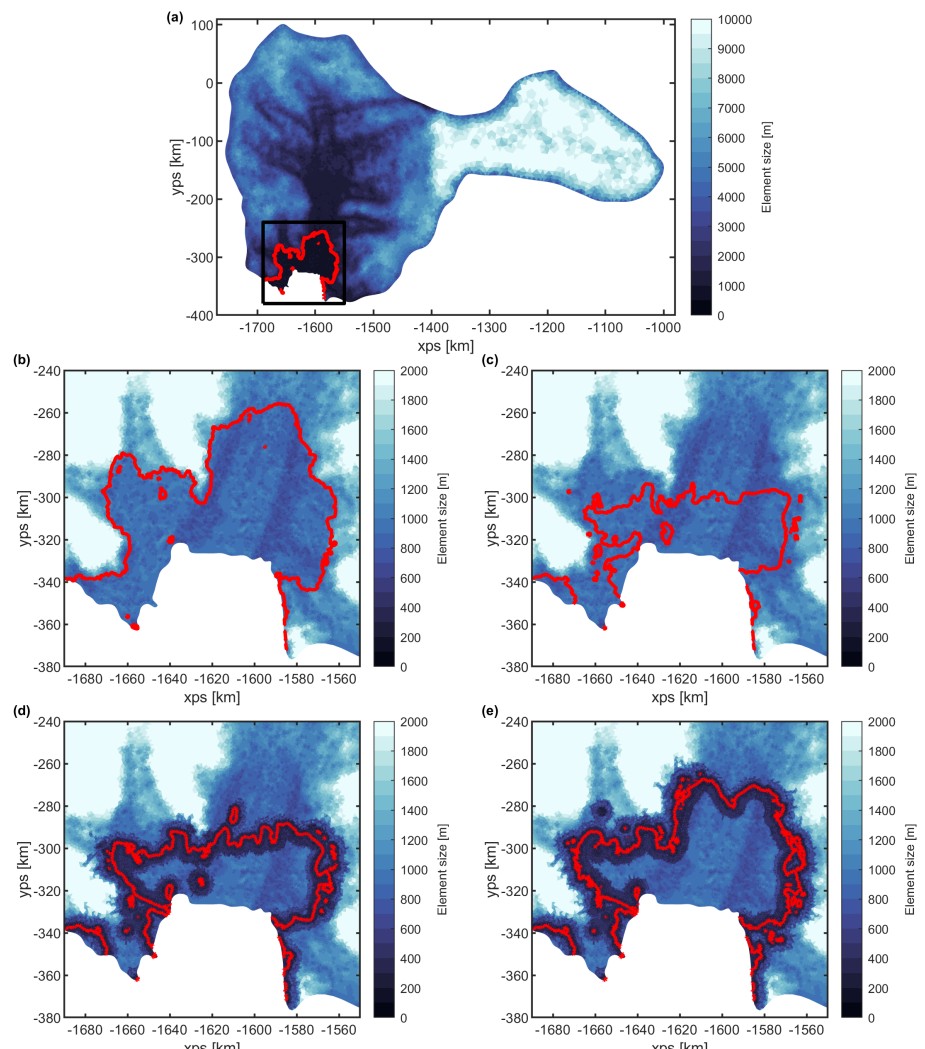

**Figure B2.** Mesh element sizes for the whole model domain **(a)** and floating ice shelf **(b)** for a present-day PIG geometry. **(c)** Element sizes for the steady-state geometry of PIG grounded at the subglacial ridge at the end of the advance stage. This is the starting configuration for the relaxation phase. Element sizes after grounding-line mesh adaption at the beginning **(d)** and end **(e)** of the WARM25 experiment. The grounding line is shown as a red line in all plots and the black box in **a** shows the location of plots **(b)**–**(e)**.

## Appendix C: Model inversion

For the inversion process Úa minimizes the cost function $J = I + R$, where

$$I = \frac{1}{2\mathcal{A}} \int \left( \frac{\boldsymbol{u} - \boldsymbol{u}_{\text{obs}}}{\boldsymbol{u}_{\text{err}}} \right)^2 d\mathcal{A} \tag{C1}$$

is the velocity misfit term and

$$R = \frac{1}{2\mathcal{A}} \int \left[ \gamma_{sA}^2 \left( \nabla \log_{10} \left( \frac{A}{\hat{A}} \right) \right)^2 + \gamma_{sC}^2 \left( \nabla \log_{10} \left( \frac{C}{\hat{C}} \right) \right)^2 \right.$$
$$\left. + \gamma_{aA}^2 \left( \log_{10} \left( \frac{A}{\hat{A}} \right) \right)^2 + \gamma_{aC}^2 \left( \log_{10} \left( \frac{C}{\hat{C}} \right) \right)^2 \right] d\mathcal{A} \tag{C2}$$

is the Tikhonov regularization term. In Eq. C1, the velocity observations ($\boldsymbol{u}_{\text{obs}}$) and measurement errors ($\boldsymbol{u}_{\text{err}}$) were from the MEaSUREs Annual Antarctic Ice Velocity Maps dataset (Mouginot et al., 2017a, b) for 2014/15 and $\mathcal{A}$ is the total domain area. In Eq. C2, $\hat{A}$ and $\hat{C}$ are prior estimates for $A$ and $C$, where we used spatially uniform values $\hat{A} = 5 \times 10^{-9}$ yr$^{-1}$kPa$^{-3}$, which corresponds to an ice temperature of -15 °C, and $\hat{C} = 1.46 \times 10^{-3}$ m yr$^{-1}$kPa$^{-3}$, which was derived from Eq. A6 using $\boldsymbol{t}_b = 80$ kPa and $\boldsymbol{u}_b = 750$ m yr$^{-1}$. An L-curve approach was initially used to find optimal values for the regularization multipliers ($\gamma_{sA}, \gamma_{sC}, \gamma_{aA}, \gamma_{aC}$) in Eq. C2. However, more smoothing was required to avoid artefacts appearing when the model was advanced forward. The spatially varying basal slipperiness ($C$) was derived through the inversion process for present-day grounded areas only. Lacking data for the 1940s period, we chose to set the downstream region to a constant value of 0.05 m yr$^{-1}$kPa$^{-3}$, which is an average value from the upstream fast flowing tributaries. Whilst a more realistic field may alter the timescales of retreat, we do not expect this to change the overall outcome of this study, as has been previously shown by Reed et al. (2023). The present-day configuration derived from the inversion and used in the next stage is shown in Fig. 1, and the inversion results are shown in Fig. C1.

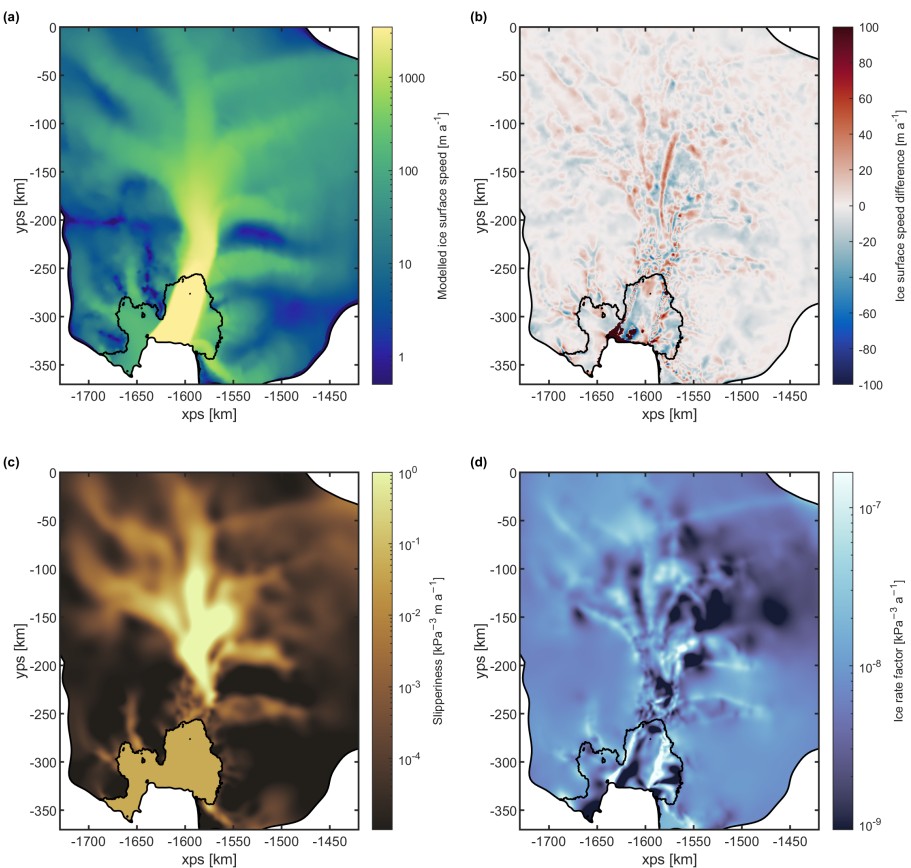

**Figure C1.** Model inversion results. **(a)** Modelled ice speed after the inversion process. **(b)** Difference between modelled and observed ice speeds (mod minus obs). **(c)** Basal slipperiness ($C$), where lighter colours indicate higher slipperiness. Downstream of the grounding line the slipperiness is set to a constant value to allow an advance to the subglacial ridge. **(d)** Rate factor ($A$) on a colour scale equivalent to depth-averaged ice temperatures from -35°C (darker colours) to 0°C (lighter colours). The model boundary and grounding line are shown as thin and thick black lines respectively. Note the log-scale colour maps for **(a)**, **(c)** and **(d)**.

## Appendix D: Flowline profiles

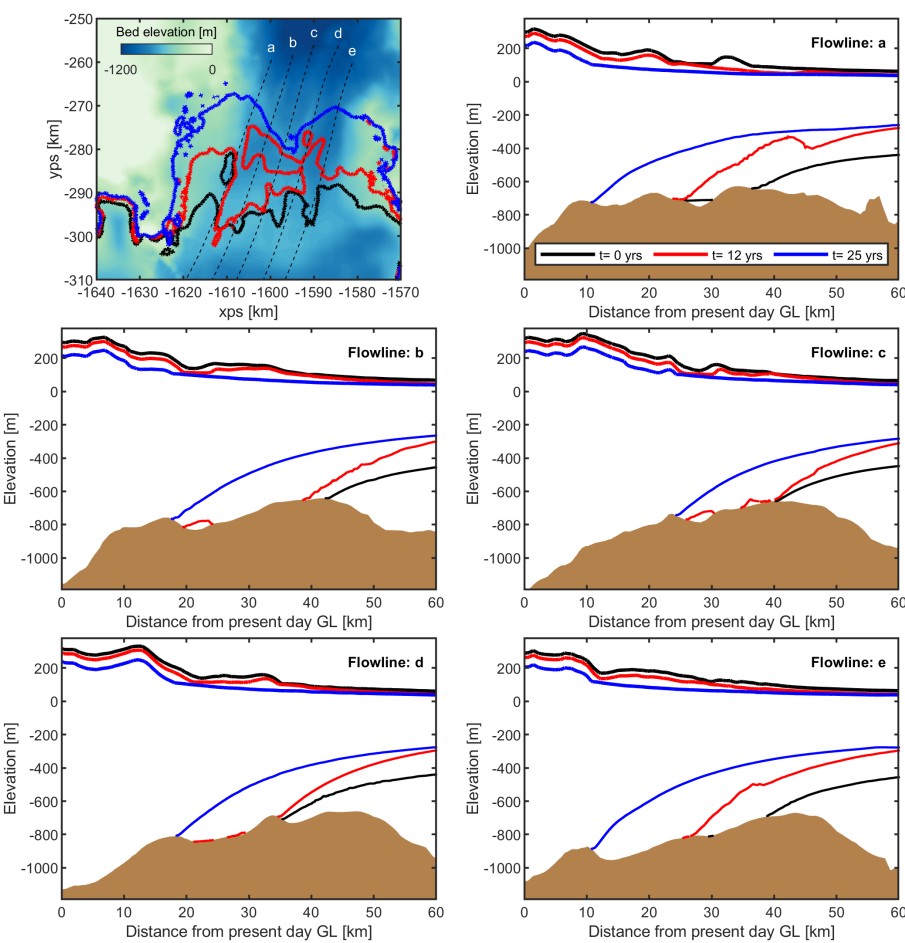

**Figure D1.** Flowline profiles after $t = 0$, 12, 25 years during the WARM25 experiment. All profile lines start at the present-day grounding-line position, as shown on the bed elevation plot.

 **Appendix E: Evolution of grounded area**

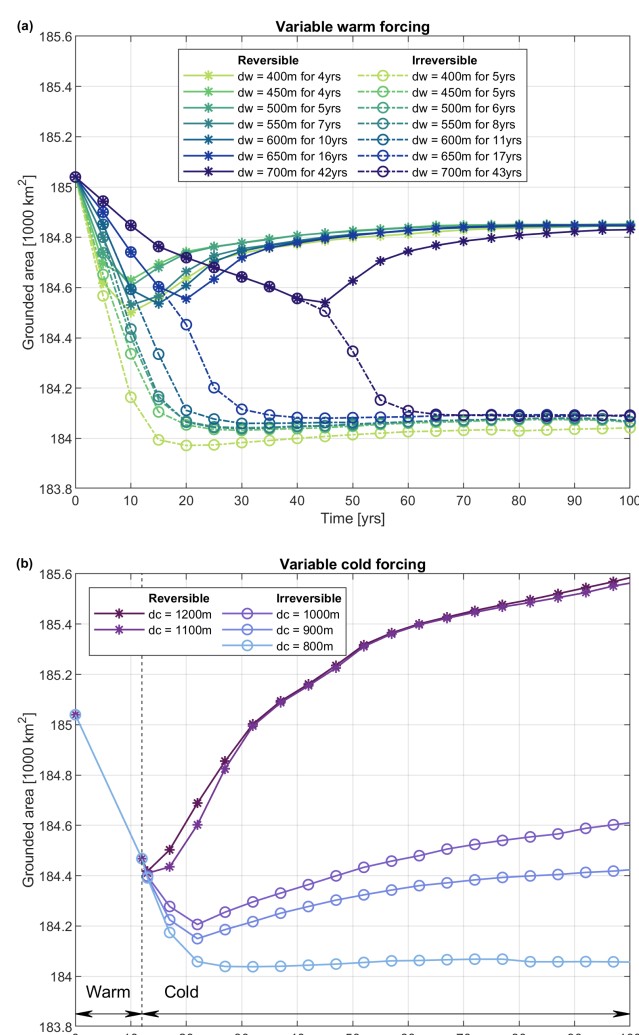

**Figure E1.** Evolution of grounded area during the variable warm forcing (top) and variable cold forcing (bottom) simulations. Star markers show reversible results, whereas open circles show irreversible results.

*Code and data availability.* Code for the open-source ice-flow model Úa is available at https://doi.org/10.5281/zenodo.3706623 (Gudmunds-son, 2020). Raw model outputs and scripts are available from the corresponding author on request and will be available as a repository on Zenodo upon publication.

*Author contributions.* All model simulations and analyses were carried out by BR, with assistance from GHG. BR wrote the manuscript and
all authors contributed to conception of the study and the editing of the manuscript.

*Competing interests.* The authors declare that they have no conflict of interest.

*Acknowledgements.* The authors acknowledge the support and resources of Supercomputing Wales, which is part-funded by the European
Regional Development Fund (ERDF) via the Welsh Government and the collaborating universities. B.R. was supported by the ENVISION
Doctoral Training Partnership studentship from the Natural Environment Research Council and by UKRI (grant number MR/W011816/1).
G.H.G and A.J. have received funding from the European Union's Horizon 2020 research and innovation programme under grant agreement
no. 820575: Tipping Points in the Antarctic Climate Components.

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
