# Peer review of "Melt sensitivity of irreversible retreat of Pine Island Glacier"

_EGUsphere, 2024_

## Author Comment (AC1)

**Melt sensitivity of irreversible retreat of Pine Island Glacier**

Brad Reed, J. A. Mattias Green, Adrian Jenkins, and G. Hilmar Gudmundsson

**Reply to referee #1 comments**

We thank the referee for their helpful and thorough feedback on our manuscript. We have responded to their comments below. The referee comment is shown in black; our replies are shown in *__bold italic blue__* and the original and new content in the manuscript is given in quoted *italic blue* *with added/changed words underlined*.

**Main comments:**

l.101: What **adjustments** where made to the ice shelf **thickness**? What part of the ice shelf was impacted? And what was the magnitude of this correction? Some **overall numbers** should be added to the text.

➢ ***The thickness adjustments were personally provided by Mathieu Morlighem (and were later incorporated into Bedmachine v3). The region affected was the ice plain area, which was grounded in the 1990s. The maximum change to ice thickness was a decrease of ~250m in a small isolated region, but typically decreases were 80-100m across the ice plain area.***

- **Original** [100-101]: *"Some adjustments were made to the ice-shelf thickness near the grounding line to ensure the hydrostatic floating condition was met for our geometry"*

- **New**: *"Some small adjustments were made to the ice-shelf thickness in the ice plain area to ensure the hydrostatic floating condition was met for the PIG ice shelf. These updated data were provided by Mathieu Morlighem and later incorporated into Bedmachine v3. The maximum change was a thickness decrease of 250m, but generally there were decreases of between 80m and 100m."*

l.104: Additional information should be added here to understand what is done to the **friction** in the part of the domain that was grounded in 1940 (and therefore a friction coefficient is needed) but is not grounded during the satellite area (and thefore cannot be inferred with observations). How was this friction chosen? Also, given there is a large uncertainty in these values, what is the impact on the results presented?

➢ ***We have added further details to Appendix C regarding this. However, our previous study showed limited impact when changing the basal slipperiness or friction law.***

- **Original** [389-392]: *"The spatially varying basal slipperiness (C) was derived for present-day grounded areas only and without data for the 1940s period we chose to set the downstream region to a constant value. Whilst a more realistic field may alter the timescales of retreat, we do not expect this to change the overall outcome of this study, as has been previously shown by Reed et al. (2023)."*

- **New**: *"The spatially varying basal slipperiness (C) was derived through the inversion process for present-day grounded areas only. Lacking data for the 1940s period, we chose to set the downstream region to a constant value of 0.05m yr$^{-1}$ kPa$^{-3}$, which is an average value from the upstream fast flowing tributaries. Whilst a more realistic field may alter the timescales of retreat, we do not expect this to have a substantial impact on our results or change the overall outcome of this study, as has been previously shown in Reed et al. (2023)."*

l.133: why was a period of **1000 years** chosen to simulated the grounding line advance? And why was the melt set-up to **zero instead of cold conditions**? I am sure a number of conditions could lead to a more or less similarly advanced position, so I am curious why such conditions were chosen?

➢ ***We used 1000 years to ensure a quasi-steady state was reached, which could then be used as a starting point for our perturbation experiments. The initial advance to the ridge is effectively finished after 100-150 years, and this is followed by only small changes to the ice shelf thickness and a small advance of the grounding line in the smaller eastern cavity.***

➢ **In our previous paper (Reed et al. 2023) we found steady-state grounding lines for a range of thermocline depths when starting from a present-day position, and showed that a steady-state at the ridge could only be achieved with a thermocline depth of at least 1100m (when using this melt parameterization), which is effectively zero melt (see Fig.4 for COLD1000 and COLD1200). As we wanted to start the simulation from an advanced position on the ridge, to agree with sediment core analysis (Smith et al., 2017), we decided to start with zero melt and then follow with a cold perturbation to ensure the ice shelf wasn't unreasonably thick. If we would have started with the cold conditions (e.g., thermocline 800m) this would have resulted in an advance only to the small bed rise ~18km downstream of the present-day position, rather than the larger ridge ~40km downstream.**

➢ **We will add this detail to the methods section.**

l.143: similarly to my question about the grounding line advance, why was a period of **100 years** chosen for the initial grounding line retreat?

➢ **The relaxation simulation, with a deep thermocline of 800m, was run to ensure there was a more realistic ice shelf draft before the warmer scenario started (otherwise a very thick ice shelf would lead to initially large melt rates). As with the advance case, 100 years was chosen to ensure a quasi-steady state was reached with this new forcing. Most of the thinning and grounding line retreat occurs within the first 10-20 years, with only small changes afterwards.**

➢ **We will add this detail to the methods section.**

**Fig.6**: the small inset showing the thermocline depth and its timing is very useful to understand the experiments and should be **provided earlier** (for example in the description of **experiments**) for improved clarity.

➢ **We will include this figure earlier in the manuscript.**

l.267: the experiments performed show this behavior for Pine Island Glacier but **not that it could happen to other places** at the same time, so this statement should be toned down.

• **Original** [265-267]: *"Our modelling results combined with observations demonstrate that glacier flow and mass loss can be sensitive to changes in ocean conditions, when grounded on a topographic high, and this can happen simultaneously across multiple glaciers."*

• **New**: *"Our modelling results combined with observations demonstrate that glacier flow and mass loss can be sensitive to changes in ocean conditions, when grounded on a topographic high, and we would expect other glaciers to respond similarly if they are in a comparable configuration."*

l.321: How does the retreat and possible readvance from this ridge compare to the present day conditions investigated in previous studies such as **Favier et al. 2014 or Seroussi et al. 2014?**

➢ **Similar to our results, in both the Favier et al. 2014 ("F14") study and Seroussi et al., 2014 ("S14") study they show a 30-40 km retreat across a retrograde slope when there is high melting at the grounding line. They also see a propagation of thinning and acceleration upstream of the grounding line, not just confined to the ice shelf.**

➢ **In the F14 study, once the grounding line retreats across the retrograde bed, the imbalance decreases, as we also see in our results. In their full Stokes simulations, a readvance is only possible when melt rates are 5-10% of the control run, or 5-10m/yr melting below the thermocline (compared to 100m/yr in the control). Similarly, we only achieve readvance once melt is reduced to less than 10%.**

➢ **Both studies suggest that a temporary increase in ice shelf basal melting can lead to acceleration and irreversible retreat, despite returning to previous conditions.**

➢ **We will include these studies in our discussion.**

**Technical comments:**

l.40: there was not much slow down reported in Mouginot et al. 2014 outside of the Eastern Thwaites ice shelf, maybe less acceleration or relatively stagnant conditions, but **not really a sector wide slowdown** and the **discharge kept acceleration at least remained constant.**

- **Original** [39-41]: *"Conversely, a deep thermocline in 2012, following a strong La Niña event in 2011, led to the lowest basal melt rates recorded in the ASE and likely caused a sector wide slow down (Mouginot et al., 2014; Dutrieux et al., 2014)."*

- **New**: *"Conversely, a deep thermocline in 2012 (800m), following a strong La Niña event in 2011, caused the lowest basal melt rates recorded in the ASE and possibly led to reduced glacier acceleration across the sector (Mouginot et al., 2014; Dutrieux et al., 2014)."*

l.83: why not use the **actual velocity at the divide** instead of zero? It is unlikely the velocity changed much during this period.

➢ **The velocity at the ice divide is small (~10s m/yr) with a high relative error (>5m/yr) and so we do not expect this to hugely impact the fast-flowing (1000s m/yr) central trunk of PIG.**

l.92: What is the **refinement**? It would be good to put an actual number to get at least the order of magnitude in the text.

- **Original** [90-92]: "*For cold and warm transient experiments (Sect. 2.6) a further time-dependent mesh refinement was applied around the grounding line, adapting the mesh as the geometry evolved, to ensure element sizes were less than 500 m in the area of transition from grounded to floating ice*"

- **New**: *"For the cold and warm transient experiments (Sect. 2.6) a further time-dependent mesh refinement was applied around the grounding line, adapting the mesh as the geometry evolved every half a year. This refinement ensured 500m mesh elements within 5000m of the grounding line, and 250m elements within 2000m of the grounding line."*

**Fig.1: the blue line** for A-B on panel 1 is hard to see

- **Original**: *"(d) Corresponding flowline profiles for the location shown in dashed blue in a, with the flow direction from A to B"*

- **New**: *"(d) Corresponding flowline profiles for the location shown as a thick black dashed line in a, with the flow direction from A to B"*

[Figure]

Fig.2 caption: Why grounding line is displayed on panel c?

- **Original**: *"Depth-dependent melt-rate parameterization for the cold (blue) and warm (red) forcing (b) and bed elevation with overlain grounding line (c). Basal melt at the start of the perturbation experiments for the cold parameterization (d) and the warm parameterization (e)."*

- **New**: *"(b) Depth-dependent melt-rate parameterization for the cold (blue) and warm (red) forcing. Bed elevation (c), basal melt for the cold parameterization (d) and basal melt for the warm parameterization (e) at the start of the perturbation experiments. In (c) – (e) the grounding line is shown as a thick black line, and model boundary as a thin black line."*

Table 1 and text lines 172-179 : it would be great to add the **total number of experiments** performed as part of the **WARMvar** and **CODLvar** cases. Were all the possible combinations tested? If not which ones were tested and how was that decided?

- **Original** [172-177]: *"The final set of experiments test the sensitivity of irreversible retreat for a wider suite of forcing conditions (WARMvar and COLDvar). All model simulations start at the ridge and consist of a period of warm forcing, followed by cold forcing. This allowed us to test whether any retreat was irreversible or not. We first experimented with the warm anomaly, by changing the duration of forcing (0 to 50 years) and the thermocline depth (400 to 700 m), where each of the experiments was followed by a 50 year period of cold forcing with an 800 m thermocline depth. The next experiment varied the cold forcing, after an initial warm anomaly, by changing the cold duration (0 to 50 years) and the thermocline depth (800 to 1200 m)"*

- **New**: *"The final set of experiments test the sensitivity of irreversible retreat for a wider suite of forcing conditions (WARMvar and COLDvar). All model simulations start at the ridge and consist of a period of warm forcing, followed by cold forcing. This allowed us to test whether any retreat was irreversible or not. We first experimented with the warm anomaly, by changing the duration of forcing (between 0 and 50 years) and the thermocline depth (400 to 700 m), where each of the warm forcing periods was followed by a 50 year period of cold forcing with an 800 m thermocline depth. There were 46 WARMvar model simulations in total, where we initially used increments of 5 years, and then narrowed this to every 1 year to find the irreversible transition year. The next experiment varied the cold forcing, after an initial warm anomaly for 12 years, by changing the cold thermocline depth (800 to 1200 m) and finding which simulation led to reversible retreat. These five simulations all ran for 100 years."*

**To include in Table 1:**

| Experiment | Warm duration $(t_w)$ [yrs] | Warm thermocline $(d_w)$ [m] | Cold duration $(t_c)$ [yrs] | Cold thermocline $(d_c)$ [m] |
|---|---|---|---|---|
| ... | ... | ... | ... | ... |
| WARMvar | 2, 3, 4, 5, 6, 10 | 400 | 50 | 800 |
| | 2, 3, 4, 5, 6, 10 | 450 | 50 | 800 |
| | 2, 4, 5, 6, 7, 8, 9, 10 | 500 | 50 | 800 |
| | 5, 6, 7, 8, 9, 10 | 550 | 50 | 800 |
| | 5, 9, 10, 11, 12, 15 | 600 | 50 | 800 |
| | 15, 16, 17, 18, 19, 20 | 650 | 50 | 800 |
| | 35, 40, 41, 42, 43, 44, 45, 50 | 700 | 50 | 800 |
| COLDvar | 12 | 600 | 100 | 800 |
| | 12 | 600 | 100 | 900 |
| | 12 | 600 | 100 | 1000 |
| | 12 | 600 | 100 | 1100 |
| | 12 | 600 | 100 | 1200 |

Fig.3: it would be good to add the **years** at the top of the corresponding columns

➢ **This has been changed:**

[Figure]

Fig.4 caption: for which experiement are the **vertical black dashed lines**?

➢ **These lines (and caption) have now been corrected.**

● **Original**: "The vertical black dashed line indicates the time of melting starting upstream of the ridge (11.3 yrs)."

● **New**: "The vertical black dashed line indicates the time of melting starting upstream of the ridge in the WARM25 experiment (12 yrs)."

l.220: "there is A continued retreat"

➢ **Will be corrected**

l.300: It looks like the glacier continues to lose mass, so what does "**stabilized**" mean in this context?

● **Original** [299-300]: *"The irreversible retreat would have been unaffected by a reverse of ocean conditions in the following years and PIG continued losing mass through the 1970s and 1980s (Jenkins et al., 2010; Mouginot et al., 2014). The glacier then stabilized when it reached an ice plain in the 1990s. This demonstrates that although an increase in basal melt is the initial cause of mass imbalance and retreat, it can be the dynamical response that becomes the dominant driver of mass loss once the forcing is removed."*

● **New**: *"The irreversible retreat would have been unaffected by a reverse of ocean conditions in the following years and PIG continued losing mass through the 1970s and 1980s (Jenkins et al., 2010; Mouginot et al., 2014). Eventually, the ice shelf detaches from the ridge and the grounding line retreats to an upstream ice plain, which leads to a reduction in ice flux across the grounding line (Mouginot et al., 2014). This sequence of events demonstrates that although an increase in basal melt is the initial cause of mass imbalance and retreat, it can be the dynamical response that becomes the dominant driver of mass loss once the forcing is removed. The glacier only stops retreating when it reaches a shallow section of bed upstream."*

---

## Author Comment (AC2)

**Melt sensitivity of irreversible retreat of Pine Island Glacier**

Brad Reed, J. A. Mattias Green, Adrian Jenkins, and G. Hilmar Gudmundsson

**Reply to referee #2 comments**

We thank the referee for their helpful and thorough feedback on our manuscript. We have responded to their comments below. The referee comment is shown in black; our replies are shown in **_bold italic blue_** and the original and new content in the manuscript is given in quoted _italic blue with added/changed words underlined_.

**Main points:**

Would results have been different if you had readvanced the grounding line under cold conditions, as used for the relaxation 100 years, rather than the no melt? (described in methods L95-100, and L130-145). Would the grounding line still have advanced, and would the steady state have been the same? Do you think the position of the steady state has any impact on the retreat i.e. would results have been any different if you'd initially advanced using cold forcing only?

➢ **In our previous paper (Reed et al. 2023) we found steady-state grounding lines for a range of thermocline depths when starting from a present-day position, and showed that a steady-state at the ridge could only be achieved with a thermocline depth of at least 1100m (when using this melt parameterization), which is effectively zero melt (see Fig.4 for COLD1000 and COLD1200). As we wanted to start the simulation from an advanced position on the ridge, to agree with sediment core analysis (Smith et al., 2017), we decided to start with zero melt and then follow with a cold perturbation to ensure the ice shelf wasn't unreasonably thick. If we would have started with the cold conditions (e.g., thermocline 800m) this would have resulted in an advance only to the small bed rise ~18km downstream of the present-day position, rather than the larger ridge ~40km downstream.**

➢ **We will include more detail in the methods section.**

I think the paper would benefit from more details/discussion of the modelling choices made in the initial state. In particular:

• There are downstream basal drag values that cannot be obtained from the initialisation because they are under ice that is floating in the present day. The authors state that these are set to a constant value (L392 and Figure C1) but this value doesn't seem to be stated. What is the value and why/how was it chosen? And how does, or might, it affect the future simulations? How can you be confident the results aren't highly dependent on this choice?

➢ **We have added further details to Appendix C regarding this. However, our previous study showed limited impact when changing the basal slipperiness or friction law.**

• **Original** [389-392]: _"The spatially varying basal slipperiness (C) was derived for present-day grounded areas only and without data for the 1940s period we chose to set the downstream region to a constant value. Whilst a more realistic field may alter the timescales of retreat, we do not expect this to change the overall outcome of this study, as has been previously shown by Reed et al. (2023)."_

• **New**: _"The spatially varying basal slipperiness (C) was derived through the inversion process for present-day grounded areas only. Lacking data for the 1940s period, we chose to set the downstream region to a constant value of 0.05m yr$^{-1}$ kPa$^{-3}$, which is an average value from the upstream fast flowing tributaries. Whilst a more realistic field may alter the timescales of retreat, we do not expect this to have a substantial impact on our results or change the overall outcome of this study, as has been previously shown in Reed et al. (2023)."_

• On L138 the authors state that the lack of advancement beyond the subglacial ridge is "aided by the fixed calving front" – can you speculate on what you'd expect if the calving front advanced? Would you expect a different steady state/more advance of the grounding line? How much of limitation is this fixed calving front for the study?

➢ **Previous studies suggest that the calving front position did not vary much between 5k years ago and 2013 (Larter et al. 2014, Arndt et al. 2018). Also, a slightly more advanced calving front compared to what we use in our study is not likely to provide much additional buttressing (Fig S4 in Fürst et al. 2016).**

➢ **We have added an additional statement to clarify this.**

- **Original** [138]: *"Hence, the subglacial ridge provides a steady-state position for PIG, which does not advance beyond it despite the absence of basal melting. This is also aided by the fixed calving front, which is not far from its 1940s position (Rignot, 2002; Arndt et al., 2018)."*

- **New**: *"Hence, the subglacial ridge provides a steady-state position for PIG, which does not advance beyond it despite the absence of basal melting. This is also aided by the fixed calving front, which is not far from its 1940s position (Rignot, 2002; Arndt et al., 2018). It is unlikely that a slightly more advanced calving front would provide much additional buttressing, (Fürst et al. 2016), so would have a limited impact on subsequent ice dynamics."*

How realistic are the thermocline heights and prescribed melt rates?

- Can you give more context, either in "2.6 Perturbation experiments" or in the Discussion, for the thermocline profiles chosen? Can you comment on how extreme are some of these e.g. the standard warm profile using in WARM25 etc, and the really high melt profiles that allows unstable retreat after just 5 or so years of forcing? (introduced in 3.4 Mapping the stability regime). It is stated that thermocline depths of 1100-1200m are unrealistic, so discussing the realistic range of warm forcings would also help, maybe around L315-320.

➢ **The thermocline depths that we chose for our warm (600m) and cold (800m) forcing are based on the shallowest and deepest observations from Pine Island Bay in 2009 and 2012/3 respectively (Dutrieux et al., 2014; Webber et al., 2017). However, as these are summer observations, we cannot rule out colder conditions (deeper thermocline) during the winter months. Furthermore, future projections for a number of possible scenarios suggest a shoaling of the thermocline with a larger volume of CDW on the continental shelf, leading to increased basal melting beneath ice shelves (Naughten et al., 2023).**

➢ **We will add more detail regarding the thermocline profiles to the "Perturbation experiments" section and in the discussion.**

- In addition, on L38-40, you mention "shallow" and "deep" thermoclines– how shallow/deep? What was the depth here compared to those explored in this study?

- **Original** [37-41]: *"A shallow thermocline in the mid to late 2000s coincided with widespread acceleration (Mouginot et al., 2014), enhanced thinning (Konrad et al., 2017) and grounding-line retreat (Rignot et al., 2014). Conversely, a deep thermocline in 2012, following a strong La 40 Niña event in 2011, led to the lowest basal melt rates recorded in the ASE and likely caused a sector wide slow down (Mouginot et al., 2014; Dutrieux et al., 2014)."*

- **New**: *"A shallow thermocline (600m) in the mid to late 2000s coincided with widespread acceleration (Mouginot et al., 2014), enhanced thinning (Konrad et al., 2017) and grounding-line retreat (Rignot et al., 2014). Conversely, a deep thermocline (800m) in 2012, following a strong La 40 Niña event in 2011, caused the lowest basal melt rates recorded in the ASE and possibly led to reduced glacier acceleration across the sector (Mouginot et al., 2014; Dutrieux et al., 2014)."*

- L117-118: thermocline depth varies, but not this melt rate, correct? Can you put this choice of 100 m/a into context here? What do you think would happen if you explored changes to this melt rate as well as thermocline depth?

➢ We chose to only vary the depth of the thermocline to be able to make direct comparisons with ocean observations/estimates (Dutrieux et al., 2014; Jenkins et al., 2018) and to be consistent with previous studies (e.g., De Rydt et al., 2014, 2016; Bradley et al. 2022).

➢ A melt rate of 100m/yr below the thermocline was based on previous estimates for this ice shelf (Bindschadler et al., 2011; Dutrieux et al., 2013; Shean et al., 2019).

➢ Another study which used a similar melt parameterization but also doubled the deep melt rate showed that this had a smaller impact compared to when the thermocline was raised (Favier et al., 2014). A modelling study of a glacier with a similar geometry has also shown that a tapering down of the melt rate close to the deep grounding lines could still lead to rapid grounding line retreat and mass loss (De Rydt et al., 2016).

➢ We will add this detail to the methods section.

**Minor points:**

L60-61: "finished when the glacier stabilized at an ice plain in the early 1990s". How do the suggest timings fit together here, when put together with L45-50: "The glacier has been retreating across an ice plain since the early 1990s" and "the subsequent ice loss was unaffected by the reduced basal melt rate in the early 2000s"? It seems from this text that the glacier stabilized in the early 1990s but was also triggered to unstably retreat at that time too?

➢ We have adjusted the wording in both places:

• **Original** [44-46]: "*In the mid to late 1990s, while the flow of most ASE glaciers had slowed down, possibly in response to cooler ocean conditions (Mouginot et al., 2014; Dutrieux et al., 2014; Naughten et al., 2022), Pine Island Glacier (PIG) continued accelerating (Rignot et al., 2002; Mouginot et al., 2014) and thinning (Shepherd et al., 2001; Wingham et al., 2009). The glacier had been retreating across an ice plain since the early 1990s (Park et al., 2013; Corr et al., 2001), where its grounding line had been situated on the seaward side of a prominent seabed rise following an earlier slow down (Mouginot et al., 2014; Jenkins et al., 2010). Although the initial cause of this recent retreat is unknown, it is clear that the subsequent mass loss was unaffected by the reduced basal melt rate in the early 2000s (Dutrieux et al., 2014).*"

• **New**: "*Between the late-1990s and mid-2000s, while most ASE glaciers experienced reduced acceleration, possibly in response to cooler ocean conditions, Pine Island Glacier (PIG) continued accelerating and thinning. The glacier had been rapidly retreating across an ice plain since the early 1990s, where its grounding line had been situated on the seaward side of a prominent seabed rise following an earlier slow down. Although the initial cause of this recent retreat is unknown, it is clear that the subsequent mass loss was unaffected by reduced basal melt rates in the early 2000s*"

• **Original** [60-61]: "*This suggests that the retreat had entered an unstable and irreversible phase after the 1940s climate anomaly, which finished when the glacier stabilized at an ice plain in the early 1990s (De Rydt and Gudmundsson, 2016; Reed et al., 2023; Mouginot et al., 2014; Park et al., 2013).*"

• **New**: "*This suggests that the retreat had entered an unstable and irreversible phase after the 1940s climate anomaly, which had finished when the glacier reached a shallower section of bed around 1990.*"

Can you deduce anything from your results about whether the unstable retreat from the 1940s can be attributed to anthropogenic change or natural variability alone? It seems to me that no trend in warming is required to sustain the retreat – quite the opposite, in fact, because in many cases the ocean has to become colder than it originally was to halt the retreat. So does this lead you to conclude that this unstable retreat could be due to natural variability alone?

➢ We know that following El Nino years there can be a significant shoaling of the thermocline in the Amundsen Sea so it's certainly possible that an increase in melting due to natural variability could have initiated PIG's retreat. However, we cannot comment on the exact cause because we have used a simplified depth-dependent melt

**parameterization here to mainly focus on the ice dynamical response. What we can say is that once the retreat started, there would need to have been a large decrease in basal melting to stop it. So, whether this was not possible because of anthropogenic change or other reasons, it is hard to say, and we will direct the reader to previous studies that have looked at the 1940s atmospheric conditions and causes in more detail (Holland et al., 2022, O'Connor et al., 2023).**

➢ **We will include this detail in the discussion.**

L119-122: it may help the reader if you link to the thermocline plots in Figure 2b here.

- **Original**: *"In the cold experiments, the shallow zero melt layer extends down to 400 m depth and the deep layer begins at 800 m depth."*

- **New**: *"In the cold experiments, the shallow zero melt layer extends down to 400 m depth and the deep layer begins at 800 m depth (Fig. 2b)."*

L138-140: The initial state from the 1000 years of no melting is "not far from the 1940s position" and after relaxation for 100 years "the new state represents the approximate situation prior to the warm anomaly in the 1940s". I'm curious how well defined the 1940s state is in Smith et al. 2017, and whether it is clear that the relaxed state matches it more closely than the unrelaxed steady state?

➢ **Just to clarify, the "not far from the 1940s position" statement is referring to the calving front position (Rignot 2002), rather than the initial state.**

➢ **In Smith et al. 2017 they deduce that there was a small cavity upstream of ridge with "space for the sediment to accumulate before 1945" but no sea water incursion. Hence, we think that our relaxed state with a thinner ice shelf and small isolated upstream cavities is probably closer to the 1940s state than the unrelaxed state.**

Figure 3, middle row: these melt rates look a bit stripey here, why is that?

➢ **This is a feature of the depth-dependent melt parameterization that we use and the way the grounding line retreats non-uniformly. Adjacent regions of shallow/deeper ice shelf draft experience different degrees of melting, which then leaves an imprint in the advected ice downstream.**

Line 200: can you comment on the timescale of the retreat here? Is it consistent with what is observed in the 1970s, or a bit slower? Would you expect it to capture the timescale?

➢ **As we only have a few observations of grounding line position (e.g., 1945, 1970s, 1992+), we do not know the exact retreat history of PIG, but those observations suggest that around 1945 the inner cavity first opened to ocean waters, and that inner cavity then remained open in the years after. The grounding line continued to retreat but the ice shelf remained in contact with the ridge until the late 1970s. So, it took around 30 years for this to occur.**

➢ **In our experiments, the inner cavity opens around 11 years and then final ungrounding from the ridge occurs approximately 8-10 years later. So, our retreat occurs at a faster rate than observed. However, we are using a simplified melt forcing which doesn't consider any geometric or topographic feedbacks, which have been shown to delay retreat by 10 years (De Rydt et al., 2016). Furthermore, we are using approximate bed conditions and ice rheology inferred from present-day velocities, so we would also expect these parameters to impact the timescale.**

Figure 4, black dashed line indicates time of melting starting upstream of the ridge – but presumably just for cases WARM12and the cold cases? Please clarify.

> **Yes, that's right, the caption has now been corrected.**

- **Original**: *"The vertical black dashed line indicates the time of melting starting upstream of the ridge (11.3 yrs)."*

- **New**: *"The vertical black dashed line indicates the time of melting starting upstream of the ridge in the WARM25 experiment (12 yrs)."*

Figure 6: can't tell that the dashed line is dashed.

> **This has now been changed to a thicker black dashed line.**

[Figure]

L290-295: you note that the stabilisation on the prograde slope might have coincided with cold ocean conditions – but would they be necessary for stabilisation in your model, **or does it stabilise there even in warm conditions?**

> **The cold conditions are not necessary for stabilisation on the prograde slope, as we see at the end of the WARM25 simulation - there is a decrease in ice flux as the grounding line stops retreating at the upstream bed rise (ice plain). In our previous paper, we also show a number of steady-state grounding lines at this location for different melt conditions.**

L332: Bett et al, 2024, also use a coupled ice-ocean model and find that ocean melt around pinning points is a key control on the retreat.

> **This reference will be added**

Figure D1: Lines for t=12 years and t=25 years hard to distinguish.

> **The line colours have been changed**

[Figure]

Figure E1: For the reversible cases the final state grows compared to the initial state – is the final grounding line position similar to the steady state or significantly more advance? Presumably not more advanced that the no melt case from the first steady state (after 1000 years of no melt)?

➢ **The final grounding line positions in the reversible cases (dc=1100 and dc=1200) are similar to the no melt (advance) case because the melt rate in both of these runs is very low.**

➢ **In Fig E1, at time=0 yrs we're showing the end of the initial relaxation stage (thermocline depth 800m), so there has already been some thinning and retreat in this case.**

---

## Author Response (AR1)

**Melt sensitivity of irreversible retreat of Pine Island Glacier**

Brad Reed, J. A. Mattias Green, Adrian Jenkins, and G. Hilmar Gudmundsson

We thank both referees for their helpful and thorough feedback on our manuscript. We have responded to their comments below. The referee comment is shown in black; our replies are shown in **bold blue** and the original and new content in the manuscript is given in quoted *italic blue*.

**Reply to referee #1 comments**

**Main comments:**

l.101: What **adjustments** where made to the ice shelf **thickness**? What part of the ice shelf was impacted? And what was the magnitude of this correction? Some **overall numbers** should be added to the text.

➢ **New**: "*Some small adjustments were made to the ice-shelf thickness in the ice plain area to ensure the hydrostatic floating condition was met for the PIG ice shelf. These updated data were provided by Mathieu Morlighem and later incorporated into Bedmachine v3. The maximum change was a thickness decrease of 250 m, but generally there were decreases of between 80 m and 100 m.*"

l.104: Additional information should be added here to understand what is done to the **friction** in the part of the domain that was grounded in 1940 (and therefore a friction coefficient is needed) but is not grounded during the satellite area (and thefore cannot be inferred with observations). How was this friction chosen? Also, given there is a large uncertainty in these values, what is the impact on the results presented?

• **We have added further details to Appendix C regarding this. However, our previous study showed limited impact when changing the basal slipperiness or friction law.**

➢ **New**: "*The spatially varying basal slipperiness (C) was derived through the inversion process for present-day grounded areas only. Lacking data for the 1940s period, we chose to set the downstream region to a constant value of 0.05 m yr−1kPa−3, which is an average value from the upstream fast flowing tributaries. Whilst a more realistic field may alter the timescales of retreat, we do not expect this to change the overall outcome of this study, as has been previously shown by Reed et al. (2023).*"

l.133: why was a period of **1000 years** chosen to simulated the grounding line advance? And why was the melt set-up to **zero instead of cold conditions**? I am sure a number of conditions could lead to a more or less similarly advanced position, so I am curious why such conditions were chosen?

• **We used 1000 years to ensure a quasi-steady state was reached, which could then be used as a starting point for our perturbation experiments. The initial advance to the ridge is effectively finished after 100-150 years, and this is followed by only small changes to the ice shelf thickness and a small advance of the grounding line in the smaller eastern cavity.**

• **In our previous paper (Reed et al. 2023) we found steady-state grounding lines for a range of thermocline depths when starting from a present-day position, and showed that a steady-state at the ridge could only be achieved with a thermocline depth of at least 1100m (when using this melt parameterization), which is effectively zero melt (see Fig.4 for COLD1000 and COLD1200). As we wanted to start the simulation from an advanced position on the ridge, to agree with sediment core analysis (Smith et al., 2017), we decided to start with zero melt and then follow with a cold perturbation to ensure the ice shelf wasn't unreasonably thick. If we would have started with the cold conditions (e.g., thermocline 800m) this would have resulted in an advance only to the small bed rise ~18km downstream of the present-day position, rather than the larger ridge ~40km downstream.**

➢ **New**: "*From the present-day configuration, we run the model with no basal melting to allow the ice stream to thicken and advance forward to the ridge. This is run for 1000 years to ensure a new quasi-steady state can be reached. Previous modelling results show that there is a steady-state position at the ridge when a deep thermocline*

*(>1000 m) is used in the melt parameterization (Reed et al., 2023), which gives close to zero melt everywhere for this geometry. Hence, we use zero melt rather than the cold conditions described in Sect. 2.4, as previous results in Reed et al. (2023) show that a 800 m deep thermocline would not be sufficient to advance from the present-day position to the ridge."*

l.143: similarly to my question about the grounding line advance, why was a period of **100 years** chosen for the initial grounding line retreat?

- **The relaxation simulation, with a deep thermocline of 800m, was run to ensure there was a more realistic ice shelf draft before the warmer scenario started (otherwise a very thick ice shelf would lead to initially large melt rates). As with the advance case, 100 years was chosen to ensure a quasi-steady state was reached with this new forcing. Most of the thinning and grounding line retreat occurs within the first 10-20 years, with only small changes afterwards.**

- **New**: *"After setting up the new steady state on the ridge we relax the ice geometry to get an approximate 1940s PIG configuration, with a more realistic ice shelf draft. This is done by running the model with the cold basal melt parameterization described in Sect. 2.4; this has a thermocline depth of 800 m, and therefore the maximum melt is deeper than the crest of the ridge. However, due to the thick ice shelf at the start of the transient simulation, this initially causes high melt rates, with a mean of 40 m yr−1 and integrated melt of 97 Gt yr−1. We run the model for 100 years which is enough time to allow the ice stream to adjust to the updated forcing and reach a new quasi-steady state with basal melting applied (Fig. 1)."*

**Fig.6**: the small inset showing the thermocline depth and its timing is very useful to understand the experiments and should be **provided earlier** (for example in the description of **experiments**) for improved clarity.

- This is now provided in Figure 3.

l.267: the experiments performed show this behavior for Pine Island Glacier but **not that it could happen to other places** at the same time, so this statement should be toned down.

- **New**: *"Our modelling results combined with observations demonstrate that glacier flow and mass loss can be sensitive to changes in ocean conditions, when grounded on a topographic high, and we would expect other glaciers to respond similarly if they are in a comparable configuration."*

l.321: How does the retreat and possible readvance from this ridge compare to the present day conditions investigated in previous studies such as **Favier et al. 2014 or Seroussi et al. 2014?**

- **New**: *"Previous modelling studies also show that thinning and acceleration can propagate upstream of the grounding line when a present-day PIG is forced by warm conditions (Favier et al., 2014; Seroussi et al., 2014)."*

- **New**: *"A similar result was also shown by Favier et al. (2014) and Seroussi et al. (2014), where a temporary increase in ice shelf basal melting leads to acceleration and irreversible retreat, despite returning to previous conditions."*

**Technical comments:**

l.40: there was not much slow down reported in Mouginot et al. 2014 outside of the Eastern Thwaites ice shelf, maybe less acceleration or relatively stagnant conditions, but **not really a sector wide slowdown** and the **discharge kept acceleration at least remained constant.**

- **New**: *"Conversely, a deep thermocline (800 m) in 2012, following a strong La Niña event in 2011, caused low basal melt rates and possibly led to reduced glacier acceleration across the sector (Mouginot et al., 2014; Dutrieux et al., 2014)."*

l.83: why not use the **actual velocity at the divide** instead of zero? It is unlikely the velocity changed much during this period.

- **The velocity at the ice divide is small (~10s m/yr) with a high relative error (>5m/yr) and so we do not expect this to hugely impact the fast-flowing (1000s m/yr) central trunk of PIG.**

l.92: What is the **refinement**? It would be good to put an actual number to get at least the order of magnitude in the text.

➤ **New**: *"For cold and warm transient experiments (Sect. 2.6) a further time-dependent mesh refinement was applied around the grounding line, adapting the mesh as the geometry evolved every half a year. This refinement ensured 500 m mesh elements within 5000 m of the grounding line, and 250 m elements within 2000 m of the grounding line (Fig. B2)."*

**Fig.1: the blue line** for A-B on panel 1 is hard to see

● This is now changed to a thick black dashed line.

➤ **New**: *"(d) Corresponding flowline profiles for the location shown as a thick dashed black line in a, with the flow direction from A to B"*

[Figure]

Fig.2 caption: Why grounding line is displayed on panel c?

➤ **New**: *"(b) Depth-dependent melt-rate parameterization for the cold (blue) and warm (red) forcing. (c) Bed elevation, (d) basal melt rate for the cold parameterization and (e) basal melt rate for the warm parameterization, at the start of the perturbation experiments. In (c) – (e) the grounding line is shown as a thick black line and model boundary as a thin black line."*

Table 1 and text lines 172-179 : it would be great to add the **total number of experiments** performed as part of the **WARMvar** and **CODLvar** cases. Were all the possible combinations tested? If not which ones were tested and how was that decided?

➤ **New**: *"The final set of experiments test the sensitivity of irreversible retreat for a wider suite of forcing conditions (WARMvar and COLDvar). All model simulations start at the ridge and consist of a period of warm forcing, followed by cold forcing. This allowed us to test whether any retreat was irreversible or not. We first experimented with the warm anomaly, by changing the duration of forcing (between 0 and 50 years) and the thermocline depth (400 to 700 m), where each of the warm perturbations was followed by a 50 year period of cold forcing with an 800 m thermocline depth. The warm forcing here spans the shallowest thermocline depths observed in Pine Island Bay (Dutrieux et al., 2014; Webber et al., 2017) and predicted under future conditions (Naughten et al., 2023). In total, there were 46 WARMvar model simulations with varying durations of warm forcing and thermocline depths. Not all combinations of parameters were tested as we were only interested in when the irreversible transition occurred.*

*The next experiment varied the cold forcing, after an initial warm anomaly, by changing the thermocline depth (800 to 1200 m) and then finding which simulation had a reversible retreat. These five simulations all ran for 100 years, and had the same initial warm forcing of a 600 m thermocline for 12 years, so that melting had already started upstream of the ridge. Although the deepest thermocline observed in Pine Island Bay was 800 m in 2012 to*

*2013, we include deeper thermoclines to account for possible cold convection events occurring earlier in the twentieth century (Naughten et al., 2022).”*

**Now included in Table 1:**

| Experiment | Warm duration $(t_w)$ [yrs] | Warm thermocline $(d_w)$ [m] | Cold duration $(t_c)$ [yrs] | Cold thermocline $(d_c)$ [m] |
|---|---|---|---|---|
| … | … | … | … | … |
| WARMvar | 2, 3, 4, 5, 6, 10 | 400 | 50 | 800 |
| | 2, 3, 4, 5, 6, 10 | 450 | 50 | 800 |
| | 2, 4, 5, 6, 7, 8, 9, 10 | 500 | 50 | 800 |
| | 5, 6, 7, 8, 9, 10 | 550 | 50 | 800 |
| | 5, 9, 10, 11, 12, 15 | 600 | 50 | 800 |
| | 15, 16, 17, 18, 19, 20 | 650 | 50 | 800 |
| | 35, 40, 41, 42, 43, 44, 45, 50 | 700 | 50 | 800 |
| COLDvar | 12 | 600 | 100 | 800 |
| | 12 | 600 | 100 | 900 |
| | 12 | 600 | 100 | 1000 |
| | 12 | 600 | 100 | 1100 |
| | 12 | 600 | 100 | 1200 |

Fig.3: it would be good to add the **years** at the top of the corresponding columns

- **This has been changed:**

[Figure]

Fig.4 caption: for which experiement are the **vertical black dashed lines**?

- **These lines (and caption) have now been corrected.**

- ➢ **New**: "The vertical black dashed line indicates the time of melting starting upstream of the ridge in the WARM25 experiment (12 yrs)."

l.220: "there is A continued retreat"

- **Now corrected**

l.300: It looks like the glacier continues to lose mass, so what does "**stabilized**" mean in this context?

- ➢ **New**: *"The irreversible retreat would have been unaffected by a reverse of ocean conditions in the following years and PIG continued losing mass through the 1970s and 1980s (Jenkins et al., 2010; Mouginot et al., 2014). Eventually, the ice shelf detaches from the ridge and the grounding line retreats to an upstream ice plain, which leads to a reduction in ice flux across the grounding line (Mouginot et al., 2014). This sequence of events demonstrates that although an increase in basal melt is the initial cause of mass imbalance and retreat, it can be the dynamical response that becomes the dominant driver of mass loss once the forcing is removed. The glacier only stops retreating when it reaches a shallow section of bed upstream."*

**Reply to referee #2 comments**

**Main points:**

Would results have been different if you had readvanced the grounding line under cold conditions, as used for the relaxation 100 years, rather than the no melt? (described in methods L95-100, and L130-145). Would the grounding line still have advanced, and would the steady state have been the same? Do you think the position of the steady state has any impact on the retreat i.e. would results have been any different if you'd initially advanced using cold forcing only?

- **In our previous paper (Reed et al. 2023) we found steady-state grounding lines for a range of thermocline depths when starting from a present-day position, and showed that a steady-state at the ridge could only be achieved with a thermocline depth of at least 1100m (when using this melt parameterization), which is effectively zero melt (see Fig.4 for COLD1000 and COLD1200). As we wanted to start the simulation from an advanced position on the ridge, to agree with sediment core analysis (Smith et al., 2017), we decided to start with zero melt and then follow with a cold perturbation to ensure the ice shelf wasn't unreasonably thick. If we would have started with the cold conditions (e.g., thermocline 800m) this would have resulted in an advance only to the small bed rise ~18km downstream of the present-day position, rather than the larger ridge ~40km downstream.**

- **New**: *"From the present-day configuration, we run the model with no basal melting to allow the ice stream to thicken and advance forward to the ridge. This is run for 1000 years to ensure a new quasi-steady state can be reached. Previous modelling results show that there is a steady-state position at the ridge when a deep thermocline (>1000 m) is used in the melt parameterization (Reed et al., 2023), which gives close to zero melt everywhere for this geometry. Hence, we use zero melt rather than the cold conditions described in Sect. 2.4, as previous results in Reed et al. (2023) show that a 800 m deep thermocline would not be sufficient to advance from the present-day position to the ridge."*

I think the paper would benefit from more details/discussion of the modelling choices made in the initial state. In particular:

- There are downstream basal drag values that cannot be obtained from the initialisation because they are under ice that is floating in the present day. The authors state that these are set to a constant value (L392 and Figure C1) but this value doesn't seem to be stated. What is the value and why/how was it chosen? And how does, or might, it affect the future simulations? How can you be confident the results aren't highly dependent on this choice?

- **We have added further details to Appendix C regarding this. However, our previous study showed limited impact when changing the basal slipperiness or friction law.**

➢ **New**: *"The spatially varying basal slipperiness (C) was derived through the inversion process for present-day grounded areas only. Lacking data for the 1940s period, we chose to set the downstream region to a constant value of 0.05 m yr−1kPa−3, which is an average value from the upstream fast flowing tributaries. Whilst a more realistic field may alter the timescales of retreat, we do not expect this to change the overall outcome of this study, as has been previously shown by Reed et al. (2023)."*

• On L138 the authors state that the lack of advancement beyond the subglacial ridge is "aided by the fixed calving front" – can you speculate on what you'd expect if the calving front advanced? Would you expect a different steady state/more advance of the grounding line? How much of limitation is this fixed calving front for the study?

• **Previous studies suggest that the calving front position did not vary much between 5k years ago and 2013 (Larter et al. 2014, Arndt et al. 2018). Also, a slightly more advanced calving front compared to what we use in our study is not likely to provide much additional buttressing (Fig S4 in Fürst et al. 2016).**

• **We have added an additional statement to clarify this.**

➢ **New**: *"Hence, the subglacial ridge provides a steady-state position for PIG, which does not advance beyond it despite the absence of basal melting. This is also aided by the fixed calving front, which is not far from its 1940s position (Rignot, 2002; Arndt et al., 2018). It is unlikely that a slightly more advanced calving front would provide much additional buttressing, (Fürst et al., 2016), so would have a limited impact on subsequent ice dynamics."*

How realistic are the thermocline heights and prescribed melt rates?

• Can you give more context, either in "2.6 Perturbation experiments" or in the Discussion, for the thermocline profiles chosen? Can you comment on how extreme are some of these e.g. the standard warm profile using in WARM25 etc, and the really high melt profiles that allows unstable retreat after just 5 or so years of forcing? (introduced in 3.4 Mapping the stability regime). It is stated that thermocline depths of 1100-1200m are unrealistic, so discussing the realistic range of warm forcings would also help, maybe around L315-320.

➢ **New**: *"In the cold experiments, the shallow zero melt layer extends down to 400 m depth and the deep layer begins at 800 m depth (Fig. 2b). We refer to this cold parameterization as having a thermocline depth of 800 m to keep consistent with previous studies (Favier et al., 2014; De Rydt and Gudmundsson, 2016; Reed et al., 2023). This forcing is based on the deepest thermocline and coldest conditions observed in Pine Island Bay between 2012 and 2013 (Dutrieux et al., 2014; Webber et al., 2017). In the warm experiments, the thermocline is shifted upwards by 200 m, so has a depth of 600 m. This is representative of the warmest conditions and shallowest thermocline recorded in Pine Island Bay in 2009 (Dutrieux et al., 2014)."*

➢ **New**: *"The warm forcing here spans the shallowest thermocline depths observed in Pine Island Bay (Dutrieux et al., 2014; Webber et al., 2017) and predicted under future conditions (Naughten et al., 2023)."*

➢ **New**: *"Although the deepest thermocline observed in Pine Island Bay was 800 m in 2012 to 2013, we include deeper thermoclines to account for possible cold convection events occurring earlier in the twentieth century (Naughten et al., 2022)."*

➢ **New**: *"The shallowest thermocline results, of 500 m and above, show that if there is a thicker layer of CDW on the shelf and therefore greater melting, as has been shown is possible in future projections (Naughten et al., 2023), it means that an unstable response of a glacier like this could be triggered in just five to six years."*

• In addition, on L38-40, you mention "shallow" and "deep" thermoclines– how shallow/deep? What was the depth here compared to those explored in this study?

➢ **New**: *"A shallow thermocline (600 m) in the mid to late 2000s coincided with widespread acceleration (Mouginot et al., 2014), enhanced thinning (Konrad et al., 2017) and grounding-line retreat (Rignot et al., 2014). Conversely, a deep thermocline (800 m) in 2012, following a strong La Niña event in 2011, caused low basal melt rates and possibly led to reduced glacier acceleration across the sector (Mouginot et al., 2014; Dutrieux et al., 2014)."*

- L117-118: thermocline depth varies, but not this melt rate, correct? Can you put this choice of 100 m/a into context here? What do you think would happen if you explored changes to this melt rate as well as thermocline depth?

➢ **New**: *"Similar to previous studies (Favier et al., 2014; De Rydt and Gudmundsson, 2016; Reed et al., 2023), the parameterization uses a piecewise-linear function of depth with zero melt in the shallow and 100 m yr−1 in the deep (Bindschadler et al., 2011; Dutrieux et al., 2013; Shean et al., 2019), and these are separated by a 400 m thick thermocline."*

➢ **New**: *"The melt rate below the thermocline is kept constant at 100 m yr−1 in all experiments, but the depth of the thermocline is varied to make direct comparisons with ocean observations (Dutrieux et al., 2014; Jenkins et al., 2018) and to be consistent with previous studies (De Rydt et al., 2014; De Rydt and Gudmundsson, 2016; Bradley et al., 2022)."*

➢ **New**: *"Furthermore, we only vary the thermocline depth and keep the melt rate constant in the deep. However, a previous study using the same parameterization showed that doubling the deep melt rate has a smaller impact compared to raising the thermocline (Favier et al., 2014). A tapering down of the melt rate at the deep grounding lines also has a limited impact on a retreating glacier that has a similar geometry to PIG (De Rydt and Gudmundsson, 2016)."*

**Minor points:**

L60-61: "finished when the glacier stabilized at an ice plain in the early 1990s". How do the suggest timings fit together here, when put together with L45-50: "The glacier has been retreating across an ice plain since the early 1990s" and "the subsequent ice loss was unaffected by the reduced basal melt rate in the early 2000s"? It seems from this text that the glacier stabilized in the early 1990s but was also triggered to unstably retreat at that time too?

- **We have adjusted the wording in both places:**

➢ **New**: "*Between the late-1990s and mid-2000s, while most ASE glaciers experienced reduced acceleration, possibly in response to cooler ocean conditions (Mouginot et al., 2014; Dutrieux et al., 2014; Naughten et al., 2022), Pine Island Glacier (PIG) continued accelerating (Rignot et al., 2002; Mouginot et al., 2014) and thinning (Shepherd et al., 2001; Wingham et al., 2009). The glacier had been retreating across an ice plain since the early 1990s (Park et al., 2013; Corr et al., 2001), where its grounding line had been situated on the seaward side of a prominent seabed rise following an earlier slow down (Mouginot et al., 2014; Jenkins et al., 2010). Although the initial cause of this recent retreat is unknown, it is clear that the subsequent mass loss was unaffected by the reduced basal melt rate in the early 2000s (Dutrieux et al., 2014)*"

➢ **New**: *"This suggests that the retreat had entered an unstable and irreversible phase after the 1940s climate anomaly, which had finished when the glacier reached a shallower section of bed around 1990"*

Can you deduce anything from your results about whether the unstable retreat from the 1940s can be attributed to anthropogenic change or natural variability alone? It seems to me that no trend in warming is required to sustain the retreat – quite the opposite, in fact, because in many cases the ocean has to become colder than it originally was to halt the retreat. So does this lead you to conclude that this unstable retreat could be due to natural variability alone?

➢ **New**: *"From our results we cannot conclude whether the unstable retreat from the ridge was caused by natural variability alone or a combination of factors (O'Connor et al., 2023), but do know that once the retreat started, it would have needed a large decrease in basal melting to overcome the ice dynamical response, and this may not have been possible because of anthropogenic change (Holland et al., 2022)."*

L119-122: it may help the reader if you link to the thermocline plots in Figure 2b here.

➢ **New**: *"In the cold experiments, the shallow zero melt layer extends down to 400 m depth and the deep layer begins at 800 m depth (Fig. 2b)."*

L138-140: The initial state from the 1000 years of no melting is "not far from the 1940s position" and after relaxation for 100 years "the new state represents the approximate situation prior to the warm anomaly in the 1940s". I'm curious how well defined the 1940s state is in Smith et al. 2017, and whether it is clear that the relaxed state matches it more closely than the unrelaxed steady state?

- **Just to clarify, the "not far from the 1940s position" statement is referring to the calving front position (Rignot 2002), rather than the initial state.**

- **In Smith et al. 2017 they deduce that there was a small cavity upstream of ridge with "space for the sediment to accumulate before 1945" but no sea water incursion. Hence, we think that our relaxed state with a thinner ice shelf and small isolated upstream cavities is probably closer to the 1940s state than the unrelaxed state.**

Figure 3, middle row: these melt rates look a bit stripey here, why is that?

- **This is a feature of the depth-dependent melt parameterization that we use and the way the grounding line retreats non-uniformly. Adjacent regions of shallow/deeper ice shelf draft experience different degrees of melting, which then leaves an imprint in the advected ice downstream.**

Line 200: can you comment on the timescale of the retreat here? Is it consistent with what is observed in the 1970s, or a bit slower? Would you expect it to capture the timescale?

➢ **New**: *"Due to the limited number of observations of grounding line position (Jenkins et al., 2010; Smith et al., 2017; Park et al., 2013), we do not know the exact retreat history of PIG. However, these observations suggest that it took approximately 30 years between the inner cavity opening to ocean waters in the 1940s and the ice shelf detaching from the ridge in the late 1970s. In our simulations this happens on a shorter timescale, of approximately 10 years. This could be due to the simplified melt forcing that we use which does not consider any geometric or topographic feedbacks that have been shown to delay retreat by 10 years (De Rydt and Gudmundsson, 2016). Furthermore, we are using approximate bed conditions and ice rheology inferred from present-day velocities, so we would also expect these parameters to impact the timescale of retreat."*

Figure 4, black dashed line indicates time of melting starting upstream of the ridge – but presumably just for cases WARM12and the cold cases? Please clarify.

- **Yes, that's right, the caption has now been corrected.**

➢ **New**: *"The vertical black dashed line indicates the time of melting starting upstream of the ridge in the WARM25 experiment (12 yrs)."*

Figure 6: can't tell that the dashed line is dashed.

- **This has now been changed to a thicker black dashed line:**

[Figure]

L290-295: you note that the stabilisation on the prograde slope might have coincided with cold ocean conditions – but would they be necessary for stabilisation in your model, **or does it stabilise there even in warm conditions?**

- **The cold conditions are not necessary for stabilisation on the prograde slope, as we see at the end of the WARM25 simulation - there is a decrease in ice flux as the grounding line stops retreating at the upstream bed rise (ice plain). In our previous paper, we also show a number of steady-state grounding lines at this location for different melt conditions.**

L332: Bett et al, 2024, also use a coupled ice-ocean model and find that ocean melt around pinning points is a key control on the retreat.

- **This reference has been added**

Figure D1: Lines for t=12 years and t=25 years hard to distinguish.

- **The line colours have been changed:**

[Figure]

Figure E1: For the reversible cases the final state grows compared to the initial state – is the final grounding line position similar to the steady state or significantly more advance? Presumably not more advanced that the no melt case from the first steady state (after 1000 years of no melt)?

- **The final grounding line positions in the reversible cases (dc=1100 and dc=1200) are similar to the no melt (advance) case because the melt rate in both of these runs is very low.**

- **In Fig E1, at time=0 yrs we're showing the end of the initial relaxation stage (thermocline depth 800m), so there has already been some thinning and retreat in this case.**